# Positive skill transfer in balance and speed control from balance bike to pedal bike in adults: A multiphase intervention study

Gary C. C. Chow[1]*, Sophia C. W. Ha[1,2]

1 Department of Health and Physical Education, The Education University of Hong Kong, Ting Kok, Hong Kong, 2 School of Health and Sports Science, Regent College London, London, United Kingdom

* ccchow@eduhk.hk

## Abstract

### Background

Learning to cycle can be challenging for adults who did not acquire the necessary skills during childhood. Balance bikes have been used to teach children how to cycle, but it was unclear whether this approach could also be effective for adults.

### Purpose

To address this, a multi-phase intervention study was conducted to investigate whether healthy adults could be taught to cycle independently through the use of a balance bike.

### Methods

In Phase 1, a case-control observational study was conducted in which 13 cyclists and 8 non-cyclists completed balance bike tests. Based on the findings, an 8-session intervention pre- and post-test study was conducted in Phase 2, using an $8 \times 20$-minute balance bike training programme to improve cycling postural stability and control. Another 11 non-cyclists completed the novel programme. The time taken to complete the balance bike tests was compared before and after the program, while their cycling confidence was recorded in each session. To assess the effectiveness of the programme, participants were invited to cycle on a pedal bike to evaluate their ability to cycle independently.

### Results

The results in Phase 1 showed that cyclists performed better on the balance bike than non-cyclists, with Bayes factor analyses providing evidence of this difference, $BF_{01} = 0.228$ in the 15 m sprint test and $BF_{01} = 0.138$ in the two-turn curved sprint test. The novel training programme in Phase 2 demonstrated remarkable effectiveness in improving their balance bike riding performance, as evidenced by the Bayes factor for completion times in the repeated measures being $BF_{01} < 0.001$. All participants were able to cycle independently with confidence after the programme.

**Data Availability Statement:** All relevant data are within the manuscript.

**Funding:** We acknowledge the support of the Seed Funding Grant 2020/21 (RG 70/2020-2021R) from the Faculty of Liberal Arts and Social Sciences, as

well as the Departmental Fund (04789) of the Department of Health and Physical Education from the Education University of Hong Kong. The funders had no role in the study design, data collection and analysis, decision to publish, or preparation of the manuscript.

**Competing interests:** The authors have declared that no competing interests exist.

## Conclusions

This study sheds light on the idea that it's never too late for adults to learn how to ride a bike. It provides evidence that healthy adults can learn to ride independently with the help of a balance bike, a tool that's commonly used for teaching children. The study identifies five key principles for effective balance bike training in adults, including focusing on riding speed, gliding to turn, building cycling confidence, engaging high motor skills, and using a dual-task approach. Our evidence-based training programme offers a safe, enjoyable, and effective way for adults to develop the skills and confidence they need to ride, even if they've never ridden before.

## 1. Introduction

Cycling is one of the most popular modes of physical activity for active transportation or leisure purposes around the world. An international study found a large increase in government support for expanding the bikeway network among 14 large bicycle-friendly cities in Europe and North America [1] to deal with the pandemic. Many cities replaced the space of motor vehicles with bike lanes for bicycles, including Paris, Berlin, and London. As a result, a healthier mode of transportation, which reduces pollution and protects citizens from viruses, was created. However, recent data noticed that an average of 9.7% of the total population in 10 countries do not know how to cycle [2].

Learning to cycle independently is a common rite of passage for children. In recent years, the average age for achieving this milestone has decreased. Children born between 2000–2019 now learn to cycle independently at an average age of 5.22 y, compared to 5.68 y for those born between 1980–1999 and 6.21 y for those born between 1960–1979 [2]. This trend can be attributed to the increasing availability of dedicated cycling tracks and child-size-friendly bicycles, which make it easier for children to learn at an earlier age. However, many individuals who struggle to cycle independently as adults often trace their difficulties back to ineffective learning approaches [3]. Negative learning experiences, such as falling while learning to ride a two-wheel bike, can discourage them from persisting in their learning journey. Similarly, using bicycles with training wheels can prove ineffective in acquiring essential cycling skills and stability [4], potentially delaying the age at which individuals learn to cycle independently [3].

In recent times, balance bikes have emerged as a popular alternative to overcome the challenges associated with traditional learning approaches for cycling. These challenges primarily involve ineffective methods and negative experiences. Balance bikes, which lack pedals and allows rider to walk or sprint while seated [5], offer a more intuitive and effective way to acquire skills and improve cycling stability. In fact, studies have shown that balance bikes are more effective in helping riders learn the essential skill of maintaining balance on two wheels compared to bikes with training wheels, which prioritize on pedalling skills [3,4]. With the right support, including access to affordable, appropriately sized adult balance bikes, along with effective learning design, individuals of all ages can overcome these barriers and learn to cycle independently.

Maintaining cycling stability mainly depends on gliding a bike with feet off the ground. [5–7]. When learning to cycle, it is not surprising that cyclists fall if they cannot maintain their posture. A balance bike is anecdotally regarded as a safe tool for learning to cycle quickly due to the similar balance strategy. When an appropriate method is used to stimulate learners' vision and their body coordination for balance, learning to cycle with minimized fear of falling

is very probable. The transfer of learning refers to the concepts that skills acquired during practice can be applied to other situations [8]. For example, if an individual learns to balance on a balance bike, they may be able to transfer that skill to riding a two-wheel bike. This is because the balance bike has helped them develop the necessary balance and coordination skills required to ride a two-wheel bike. Positive transfer occurs when the skills learned on the balance bike improve the individual's ability to a new but similar task, riding on a two-wheel bike [3]. On the other hand, negative transfer occurs if the skills learned on the bike with training wheels actually hinder the individual's ability to learn [3,6]. Zero transfer would occur if the skills learned on a balance bike have no effect on their ability to ride a pedal bike.

Riding on a balance bike has been proven to be an effective method for positively transferring the skills needed to learn how to cycle [6]. In fact, a study indicated that children could learn how to cycle within a short period of two weeks with the assistance of balance bike [7]. Additionally, a retrospective study has highlighted the benefits of using balance bike can be beneficial in educating or re-educating children to learn how to cycle, ultimately lowering the learning age [3]. Learning to control a balance bike is attainable, and even preschool children with varying fundamental motor skill level could improve the capacity of balance bike control with eight weeks of free play on a balance bike [9]. Moreover, a higher volume of practice, known as motor skill engagement (MSE), on a balance bike led to more improvement in balance bike skills [9].

However, the effectiveness of learning to cycle by using a balance bike is still a topic yet to be fully explored, and scientific evidence supporting its effectiveness is limited. If the ability to ride a balance bike is transferrable to a pedal bike, it is expected that cyclists should be superior in riding and controlling a balance bike compared with the non-cyclists. Nonetheless, it is reasonable that adults who cannot cycle independently could improve their ability through sufficient practice on a balance bike, which in turn allows them to ride a pedal bike afterwards. Additionally, it is reasonable to assume that adults may feel more confident riding a two-wheel bike becoming acquainted with riding a balance bike. Therefore, this study adopted a multi-phase intervention approach that had two aims and phases: (1) to identify whether previous cycling experience is associated with riding performance on a balance bike via a case-control observational study and develop an eight 20-minute balance bike training programme using these data; (2) to evaluate whether healthy adults without cycling experience could develop essential cycling stability on a balance bike and transfer the skills to a pedal bike using a single-arm trial.

## 2. Methods

This study was a multi-phase intervention study including a case-control observational study in Phase 1 and an $8 \times 20$-minute balance bike training single-arm trial in Phase 2 (Fig 1). From June 2021 to August 2022, healthy adults were recruited on the basis of cycling experience (with or without) by convenience sampling (via online and poster advertising). The inclusion criterion is (1) aged 18 years or above. The exclusion criteria are (1) suffering from serious injuries that may affect balance performance (previous injuries that have been fully recovered are acceptable), (2) significant musculoskeletal, cardiovascular (e.g., hypertension), neurological (e.g., peripheral neuropathy), visual, vestibular, or other sensorimotor disorders, (3) muscle fatigue on the day of the assessment, and (4) capable to ride a motorcycle. All participants were screened by a sports scientist according to the above inclusion and exclusion criteria. The eligibility criteria were the same in Phases 1 and 2.

All procedures were approved by the Human Research Ethics Committee of The Education University of Hong Kong (Ref. no. 2020-2021-0285) and in accordance with the Declaration

## Phase 1: Case-Control Observational Study

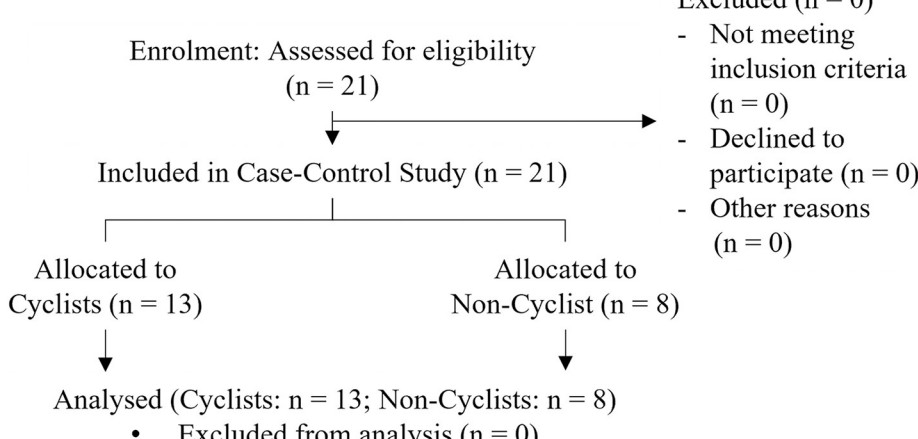

Enrolment: Assessed for eligibility
(n = 21)

Included in Case-Control Study (n = 21)

Excluded (n = 0)
- Not meeting inclusion criteria (n = 0)
- Declined to participate (n = 0)
- Other reasons (n = 0)

Allocated to Cyclists (n = 13)

Allocated to Non-Cyclist (n = 8)

Analysed (Cyclists: n = 13; Non-Cyclists: n = 8)
• Excluded from analysis (n = 0)

---

### Intervention Development
### An 8 × 20-minute balance bike training programme based on data collected

---

## Phase 2: Single-Arm Trial

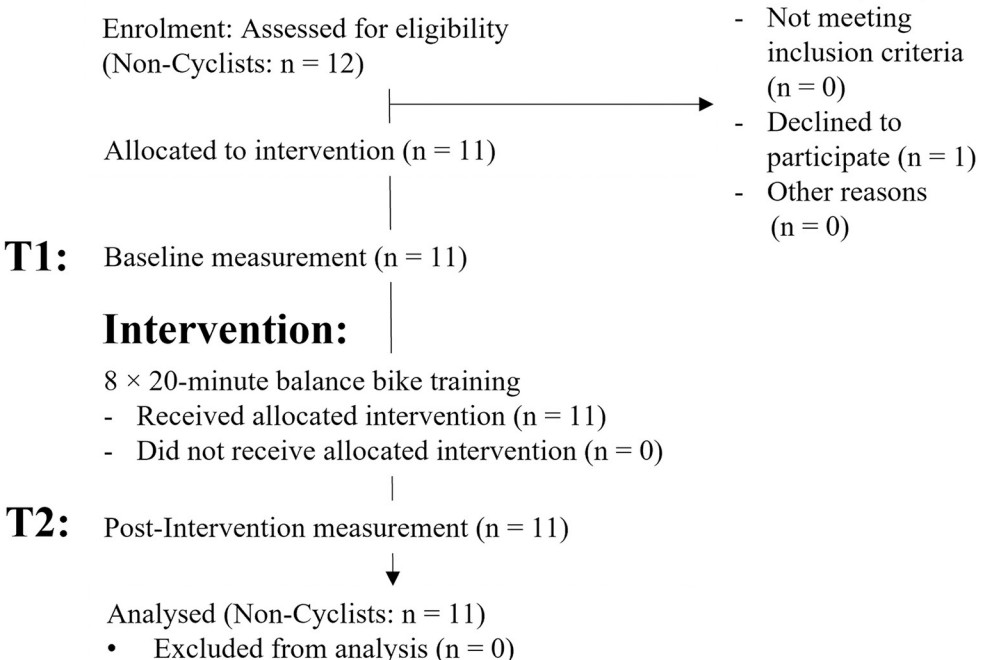

Enrolment: Assessed for eligibility
(Non-Cyclists: n = 12)

Allocated to intervention (n = 11)

Excluded (n = 0)
- Not meeting inclusion criteria (n = 0)
- Declined to participate (n = 1)
- Other reasons (n = 0)

**T1:** Baseline measurement (n = 11)

### Intervention:

8 × 20-minute balance bike training
- Received allocated intervention (n = 11)
- Did not receive allocated intervention (n = 0)

**T2:** Post-Intervention measurement (n = 11)

Analysed (Non-Cyclists: n = 11)
• Excluded from analysis (n = 0)

**Fig 1. Flowchart of this multi-phase intervention study.**

of Helsinki. Informed written consent was obtained from each participant before experimentation. The confidentiality of participants' information was ensured both during and after the data collection process. Additionally, all measurements were carried out by an experienced sports scientist, with the support of a research assistant in an indoor multi-game court to ensure accuracy and consistency. The STROBE checklist is available to the reader as S1 Appendix.

## Study design

In this study, participants were classified into two groups: cyclist group and non-cyclist group. Studies typically identified three criteria for independent cycling: the ability to self-launch, riding a bike continuously for a certain distance, and braking without assistance [6]. However, the combination of criteria varied across studies. Since braking using the handlebar is not part of a balance bike's design, participants in this study who were able to self-launch and ride a pedal bike for 5 minutes continuously were classified as the cyclist group, while those who were unable to do that were classified as the non-cyclist group.

A total of 21 participants took part in Phase 1, including 13 cyclists (6 males and 7 females; age: 22.62 ± 2.14 years; height: 167.12 ± 10.97 cm; weight: 62.49 ± 12.21 kg) and 8 non-cyclists (3 males and 5 females; age: 22.13 ± 1.36 years; height: 164.00 ± 5.45 cm; weight: 56.30 ± 3.49 kg). Their demographics (i.e., age and gender), anthropometrics (i.e., height and weight), and physical functional capacity (i.e., jump height of countermovement jump [CMJ]) were measured. Following that, the times of completion for two assigned balance bike manoeuvres were recorded. The research team used the information collected in Phase 1 to develop an eight 20-minute balance bike training programme that attempted to address non-cyclists' needs on acquiring riding skills on a balance bike.

In Phase 2, another 12 non-cyclists were recruited using the same criteria to participate in the study, of whom 11 completed the entire study (2 males and 9 females; age: 36.73 ± 15.74 years; height: 160.68 ± 9.12 cm; weight: 57.50 ± 12.57 kg). One individual withdrew from the study due to personal reasons. The 11 non-cyclists were required to attend two testing sessions at the same time on two different days (i.e., baseline, T1; post-test, T2), and each session lasted for approximately 1.5 hours. The measurements and the two balance bike manoeuvres were duplicated at T1. Participants were then encouraged to complete the 8 × 20-minute balance bike training. Their riding distance in each session and their cycling confidence level (on a pedal bike) after each session were documented. At T2, after undergoing an identical physical functional capacity test, participants were verbally interviewed about their perception of the learning experience. They were invited to self-launch and ride a pedal bike to determine whether they could be up-skilled as a cyclist after the balance bike training.

Based on the available literature, the following hypotheses have been derived concerning cycling experience and the effectiveness of the balance bike training intervention: Hypothesis 1: It is anticipated that individuals with prior cycling experience, referred to as cyclists, will display superior performance in riding and controlling a balance bike compared to individuals without prior cycling experience, referred to as non-cyclists. Hypothesis 2: Non-cyclists who undergo the balance bike training intervention will demonstrate enhanced performance in balance bike tests and experience an increase in confidence levels when riding a two-wheel bike after engaging in practice with a balance bike.

## Anthropometrics

Participants' body height and weight were measured by a portable height rod stadiometer (Detecto, PHR, MO, USA) and digital scale (Tanita BWB-800, MA, USA) respectively. They

were asked to stand on the height rod stadiometer and digital scale for the measurements. The graduation unit was 0.1 cm for height and 0.1 kg for weight.

## Physical functional capacity

Participants' physical functional capacity was measured by CMJ using Takei Vertical Jump Meter (TVJM; T.K.K. 5406 Jump MD, Takei Scientific Instruments Co. Ltd., Niigata, Japan). High reliability ($r = 0.90$) was reported for CMJ [10–13]. Participants were required to stand on the Takei rubber mat with a belt secured around the waist. The belt weighs 0.6 kg and is connected to the mat with a measurement cord which initially represented the distance between the waist and the ground. The cord elongated during a jump. The smallest measurement unit was 1 cm, and its error was up to ± 2 cm. After familiarisation, participants were instructed to jump with their hands on their hips as high as possible. The average jump height of three trials was used for analysis.

## Balance bike testing

A 15-m sprint (Fig 2A) and a two-turn curved sprint (Fig 2B) [14] were adopted from a recent study. The first manoeuvre consisted of a start, an approach to top speed, and a run-stop in two routes: an approach run, followed by a zip-zap turn. Participants were requested to ride and sprint at full speed. Sufficient time was given to participants to familiarise themselves with a 20-inch balance bike and the testing setup. To ensure that participants could comfortably and safely balance on the bike during testing, they were ample time to become familiar with a 20-inch balance bike and the testing setup. The seat height of the bike was then adjusted to fit each participant's body, enabling them to place both feet on the ground with slightly bent hips and knees [9,14]. This position allowed for proper extension of the hip and knee joints when propelling themselves forward. The average time completion (s) of five trials were assessed by a pair of timing gates (Brower Timing Systems, Utah, USA) that were placed at the start and finish lines. The measuring system has been found to be accurate in recording 0–20 m running speed ($r = 0.911$, $p < .01$) [15].

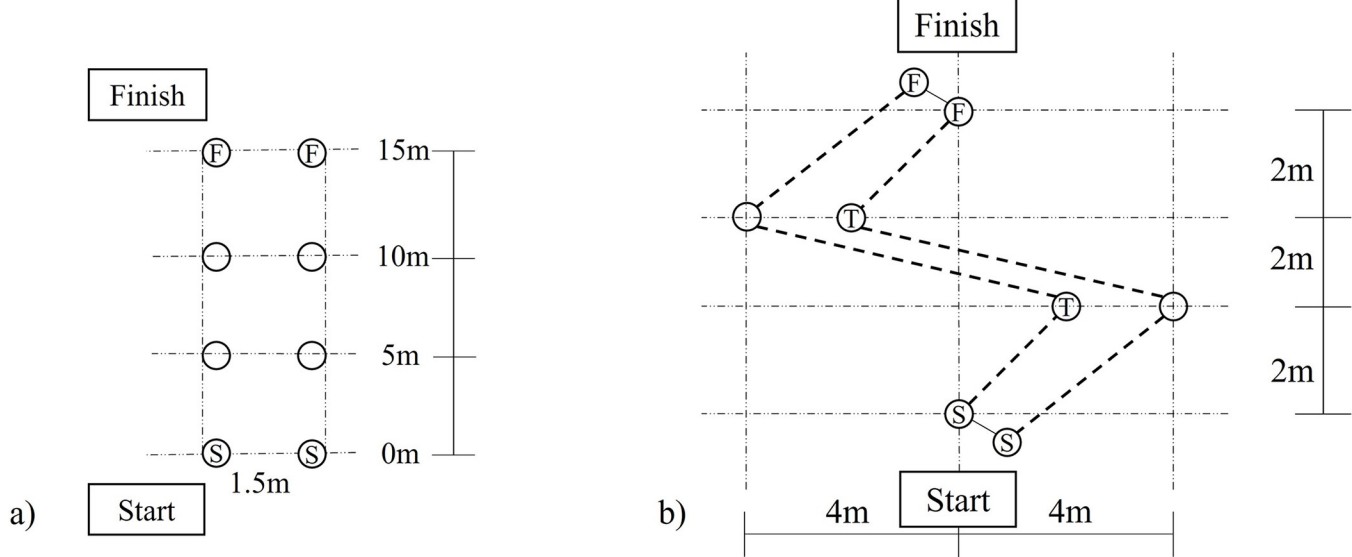

**Fig 2.** Balance bike manoeuvre setups: a) 15m sprint; b) Two-turn curved track. Note: Ⓢ: Start with timing gates; Ⓕ: Finish with timing gates; Ⓣ: Cones to indicate turning; ◯: Cones.

## Cycling confidence

Previous studies have used self-report cycling confidence to examine the effectiveness of cycling training among adults [16,17]. In this study, cycling confidence was assessed verbally after each of the eight training sessions and the two testing sessions (i.e., T1 & T2), resulting in a total of 10 recorded confidence scores. Participants were asked to rate their confidence to ride a pedal bike on a five-point Likert scale (1 = not confident; 2 = slightly confident; 3 = somewhat confident; 4 = fairly confident; 5 = very confident).

## Self-perception of the learning experience

Before the pedal bike trial at T2, a structured interview was conducted to assess the participants' perception regarding their acquisition of cycling skills during the programme. The interview lasted for 3 minutes and consisted of two questions: Q1) How did you feel after completing the balance bike training programme? and Q2) Which specific elements do you feel you developed after completing the balance bike training programme that contributed to changes of your cycling confidence?

## Cycle independently

In this study, being able to cycle independently was defined as when a participant can self-launch and ride a pedal bike for 5 minutes continuously. At the end of the balance bike training programme, participants were invited to ride a 20-inch conventional pedal bike, which was of the same size as the balance bike used in the training. The time taken by participants to cycle independently was recorded and reported, in 15-minute intervals: below 15 minutes, 16 to 30 minutes, 30 to 45 minutes, and 45 to 60 minutes. The trial had a maximum time limit of 60 minutes.

## Balance bike training programme development

In Phase 1, key cycling performance variables, issues, and preferences were determined by combing participants' functional capacity, balance bike testing data, perceived confidence in two groups. Other factors, such as cycle patterns on balance bike, previous training experience, appropriate training features and designs, and offering autonomy support in the training, were also considered in the development of balance bike training programme. After completing data collection in Phase 1, a sport scientist developed the balance bike training programme to address the needs of non-cyclists.

## Motor-skill engagement (MSE) on a balance bike

The distance travelled (km) by each participant on balance bike was collected using cycle computer (CC-RS200W, Cateye Quick Cyclocomputer, Osaka, Japan). Participants were able to read the data monitor attached to the handlebar. The data was also used to determine the MSE on a balance bike during the training programme in Phase 2.

**Data analysis.** The SPSS Statistics 28.0 software package (Armonk, NY: IBM Corp.) was used for the statistical analyses. A fully Bayesian inferential statistical approach was used to provide probabilistic statements [18]. Analysis in Phase 1, due to the lack of strong a priori evidence, non-informative prior was used [19]. A Bayesian two-sample t-tests were used to compare the outcome variables between cyclists and non-cyclists. Then, new prior distributions of cyclists and non-cyclists on the outcome parameters were collected. Specific to the non-cyclists, Bayesian related sample normal tests were used to evaluate the effects of balance bike training on the time of completion of the two balance bike tests and the jump height in CMJ.

A Bayes factor for related-sample *t* test was used to determine the changes of cycling confidence at post-test (T2) compared to the pre-test level (T1).

The strength of the evidence for the null hypothesis against the alternative hypothesis (or models) can be expressed as a Bayes Factor ($BF_{01}$), which is an odds ratio. The size of BFs can be interpreted as providing anecdotal ($BF_{01} = 1/3$–1), moderate ($BF_{01} = 1/10$–1/3), strong ($BF_{01} = 1/30$–1/10), very strong ($BF_{01} = 1/100$–1/30), and extreme ($BF_{01} < 1/100$), in favour of alternative hypothesis compared to null hypothesis [20,21]. Evidence for the null hypothesis was set as anecdotal ($BF_{01} = 1$–3), moderate ($BF_{01} = 3$–10), strong ($BF_{01} = 10$–30), very strong ($BF_{01} = 30$–100), and extreme ($BF_{01} > 100$) [20,21]. Effect size (ES), and Cohen's *d* values of 0.2, 0.6, and 1.2 were defined as small, moderate, and large effect sizes, respectively [22].

This study adopted Bayesian updating with a simulation-based approach for sample size calculation in Phase 1, sample size was continued to increase until the Bayes factor was sufficient, indicating strong evidence in favour of the alternative hypothesis.

## 3. Results

### Phase 1: Assessments between cyclists and non-cyclists

Cyclists demonstrated faster completion time on a balance bike in the 15m sprint test ($BF_{01} = 0.228$, $p = 0.013$, $d = 1.89$) and two-turn curved sprint ($BF_{01} = 0.138$, $p = 0.037$, $d = 3.31$) than the non-cyclists (Table 1). No differences were found between the two groups in the demographic data as well as the jump height in CMJ.

### Phase 1: Balance bike training programme development

The balance bike programme was based on five principles: riding speed, gliding to turn, cycling confidence, high MSE, and a dual-task approach. Each session was tailored to participants' riding abilities, and an instructor-to-participant ratio of 1: 2 to 1: 4 was used regarding the participants' availability. The training approach used an authentic learning approach that presented challenges resembling real-world scenarios. Table 2 provides an overview of the balance bike intervention framework.

Given faster times of completion of 15m sprint, and two-turn curved sprint in cyclist, the programme emphasized riding speed and gliding to turn throughout the programme. Participants were encouraged to practice the route as much as possible during the standardized

**Table 1. Balance bike performance and jump height in Phase 1 and Phase 2.**

| Phase 1: Case-Control Observational Study | | | | | |
|---|---|---|---|---|---|
| | Cyclists (n = 13) | Non-cyclists (n = 8) | Bayes Factor ($BF_{01}$) | *p* value | Effect Size (*d*) (95%CI) |
| 15m sprint test (s) | 4.90 ± 1.16 | 6.79 ± 2.03 | 0.228 | 0.013* | -1.23 (-2.18, -0.26) |
| Two-turn curved sprint test (s) | 6.35 ± 1.30 | 9.66 ± 3.61 | 0.138 | 0.007* | -1.37 (-2.33, -0.37) |
| Jump height in CMJ (cm) | 37.64 ± 8.06 | 35.13 ± 9.07 | 2.701 | 0.516 | 0.30 (-0.59, 1.18) |

| Phase 2: 8 × 20-minute balance bike training single-arm trial | | | | | |
|---|---|---|---|---|---|
| | Non-cyclists (n = 11) | | Pre–Post (95%CI) | Bayes Factor ($BF_{01}$) | *p* value | Effect Size (*d*) (95%CI) |
| | Pre | Post | | | | |
| 15m sprint test (s) | 7.07 ± 1.21 | 5.39 ± 0.92 | 1.68 (1.00, 2.36) | < 0.001 | <0.001* | 1.66 (0.71, 2.57) |
| Two-turn curved sprint test (s) | 9.25 ± 1.26 | 6.64 ± 0.83 | 2.60 (1.84, 3.37) | < 0.001 | <0.001* | 2.29 (1.13, 3.43) |
| Jump height in CMJ (cm) | 26.15 ± 8.45 | 28.42 ± 8.45 | -2.27 (-3.34, -1.20) | 0.019 | <0.001* | -1.42 (-2.26, -0.55) |

Note: CMJ: Countermovement jump; Means ± standard deviations are presented unless specified otherwise. * $p < 0.05$.

**Table 2. Brief description of 8 × 20-minute balance bike training programme.**

| Session | Prerequisite (Minimum) | Learning Objectives | | Route Design | Dual-Task Activities (e.g.) |
|---|---|---|---|---|---|
| | | Motor | Cognitive | | |
| | Able to: | Able to: | Able to: | | |
| 1st | • Walk / Run with the bike. | • Glide the bike independently.<br>• Slow down by TWO feet. | • Keep riding on slow lane.<br>• Rest / Stop only at Rest Zone.<br>• Leave the rest zone with cautious. | 4-Corners Round Route with Rest Zone (Two-Lane-One-Way) | • Talk with Others.<br>• Ride with partner side by side. |
| 2nd | • Glide the bike.<br>• Stop at rest zone when necessary. | • Glide the bike fast for 10 m.<br>• Speed up and overtake others at the fast lane.<br>• drop back if necessary and return to the slow lane.<br>• Slow down by TWO feet at high speed. | • Sound the horn before overtaking.<br>• Stay at slow lane with same speed / slow down allowing others to overtake.<br>• Keep a healthy vision by head movement for road safety | 4-Corners Round Route with Rest Zone (Two-Lane-One-Way)<br>+<br>Diagonal Highway (One-Lane-One-Way) | React to the signals by instructor.<br>• Name the object.<br>• Ride to the site. |
| 3rd | • Glide the bike fast for 10 m.<br>• Stop at rest zone when necessary.<br>• Overtake others. | • Keep your head and eyes oriented 3–4 seconds ahead. | • Turn Left / Right with early signals (e.g., hands).<br>• Enter to and exit from roundabout with clear route map in mind. | 4-Corners Round Route with Rest Zone (Two-lane-Two-way)<br>+<br>Roundabout in centre (4-Exit) | • Lead others to follow a bike trip.<br>• Follow the leaders to ride. |
| 4th | • Keep your head and eyes oriented 3–4 seconds ahead.<br>• Turn Left / Right with early signals (Voice / hands). | • Keep your head and eyes oriented 3–4 seconds ahead. | • Turn Left / Right with early signals (e.g., hands).<br>• Enter to and exit from roundabout with clear route map in mind. | 4-Corners Round Route with Rest Zone (Two-lane-Two-way)<br>+<br>Roundabout in centre (4-Exit) | Eye on the traffic chaos (Instructor play as a passenger or another rider) |
| 5th | • Control the bike with all essential skills on a wide lane. | • Lean forward and stride for sprints.<br>• Speed up and slow down for multiple turns. | • Keep on a same lane after U-turn. | E-shaped Route (Two-Lane-One-Way) | Speed Competition |
| 6th | • Control the bike with all essential skills on a wide lane. | Hand off from handle to collect and place an object when riding on a bike. | • Locate a moving object (Instructor) when riding on a bike. | E-shaped Route (Two-Lane-One-Way)<br>+<br>Collection Point | Deliver object from Collection Point to Instructor or vice versa. |
| 7th | • Control the bike with all essential skills on a wide lane. | • Maintain balance when riding on a narrow track.<br>• Maintain balance on a Two-Lane-Two-Way route. | • Decide the best option of route for task. | 4-Corners Round Route with Rest Zone (Two-lane-Two-way)<br>+<br>Diagonal Highway (One-Lane-One-Way & Narrow)<br>+<br>Collection Point | Deliver object from Collection Point to Instructor or vice versa.<br>React unexpected situation (provided by instructor) |
| 8th | • Control the bike with all essential skills on a wide route. | • Combine all skills towards both predicted and unexpected situations. | • Combine all skills towards both predicted and unexpected situations. | 4-Corners Round Route with Rest Zone (Two-lane-Two-way)<br>+<br>Roundabout in centre (4-Exit)<br>+<br>Collection Point | Deliver object from Collection Point to Instructor or vice versa.<br>Open route for free ride (No restriction on speed, direction, task). |

Note: Programme were developed based on 5 training principles for learning to cycle 1) riding speed, 2) gliding to turn, 3) cycling confidence, 4) high MSE, and 5) dual-task approach; Each training should travel in between 1.5–2 km to achieve motor skill engagement; Verbal encouragement is provided by Instructor.

20-minute session to achieve a longer travelling distance or more successful completion of task. At the end of each session, the instructor shared the distance travelled by each participant using cycle computer, allowing participants to track their progress over time.

Due to the previous poor cycling experience of adult non-cyclist, autonomy support and mastery of the skills were the key variable for their active participation (i.e., high MSE) with confidence, participants had the freedom to choose the speed, skill, and level of participation, including when to stop at rest area or use the overtaking lane. The instructor encouraged participants to ride as fast as they could, but also provided acceptance to participants who preferred to ride at their own pace, even if it was slower, with the aim of ensuring that participants felt comfortable and in control of the bike while also pushing themselves to improve their performance.

To challenge participants' cycling stability, the training programme included activities that involved riding a balance bike while performing additional tasks (i.e., dual-task approach) that required executive functions [23], such as visual scanning to find a moving target or delivering an object with one hand when riding. The programme was designed as a series of challenges that simulated different scenarios encountered on cycling tracks (Fig 3), such as navigating traffic chaos or taking turns leading the group, to encourage participants to adapt to unexpected situations for better cycling stability.

## Phase 2: Assessment for balance bike intervention

Compared to Phase 1, non-cyclists in Phase 2 (n = 11) displayed comparable demographics but older (36.72 ± 15.74 years, $BF_{01} = 0.30$, $p = 0.019$, $d = 1.21$) and jumped lower in CMJ (26.15 ± 8.45 cm, $BF_{01} = 0.53$, $p = 0.041$, $d = -1.03$). Balance bike performances in the two testing manoeuvres were similar across Phases 1 and 2.

Repeated measures were conducted for the 11 non-cyclists who attended all balance bike training and the two testing sessions (Table 2). Participants accumulated 14.88 ± 2.09 km in the eight 20-minute balance bike training sessions, which is expected to have a high MSE according to Kavanagh et al.'s [9] definition. On average, participants travelled 1.62 ± 0.40 km in the first session and gradually increased across the programme and attained 2.02 ± 0.31 km in the last session (Fig 4). As a result, spending less time at rest and/or faster riding speed were shown.

Based on the verbal survey conducted during Phase 2 of the study, participants reported feeling fairly to very high level of confidence with their cycling abilities, with a score of 4.45 ± 0.52 out of possible 5, after completing all training sessions. Participants commented that after completing the balance training programme (Q1), they felt comfortable riding a balance bike (n = 11), experienced a reduction in their fear of falling (n = 6), and felt more relaxed while holding onto the handles of the bike (n = 4). Specifically, in response to Q2, participants commented on the improvements in their cycling confidence, including an increased capacity of controlling the bike during gliding and turning (n = 7), and an increase in their skills and knowledge for handling the traffic chaos (n = 4). At the end of the study (T2), there was a significant increase in participants' cycling confidence compared to that at T1 with a difference of 3.09 ± 0.83 ($BF_{01} < 0.001$, $p < 0.001$, $d = 3.72$). Out of the eleven participants, eight were able to self-launch and ride a pedal bike continuously within 15 minutes, while the remaining three were able to cycle independently within 16 to 30 minutes (n = 1) and 45 to 60 minutes (n = 2).

## 4. Discussion

In this multi-phase intervention study, we first examined whether the riding performance on a balance bike in healthy adults with cycling skills differed from those without cycling skills (Phase 1). We then tested our novel balance bike training to investigate if it could improve the balanced bike performance of adults who could not ride independently and thus allow them to

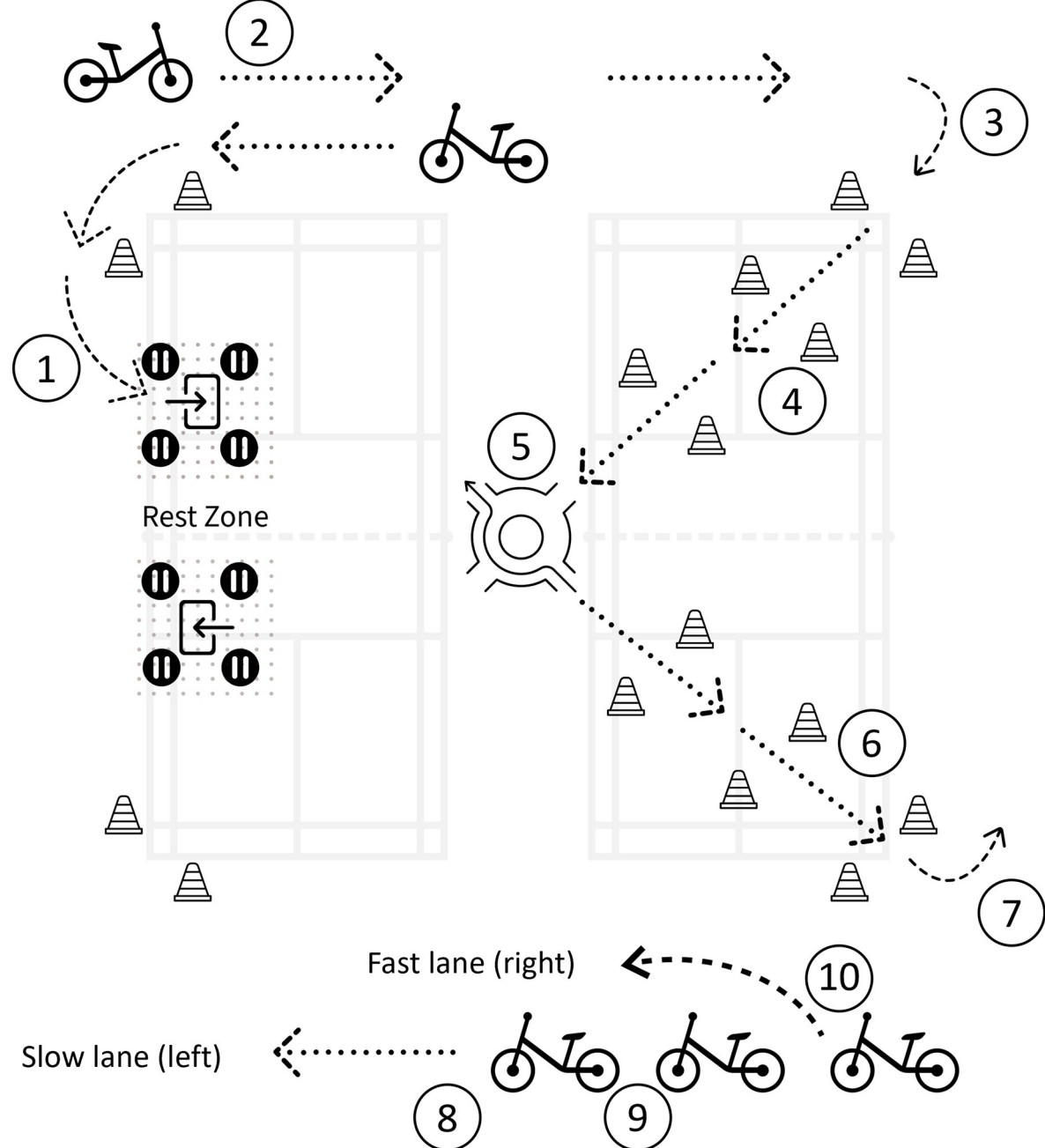

**Fig 3. A schematic image illustrating the common setting of the challenge course.** Note: (1): Stop at rest zone; (2): Encountering other riders; (3): Turn into the diagonal highway (One-Lane-One-Way); (4): Speed up at the diagonal highway; (5) Enter to and exit from roundabout in centre (4-Exit); (6) slow down at the diagonal highway before exit; (7): Exit to the major cycling track; (8) Taking turns leading the group; (9) Follow the leaders to ride; (10): Overtaking others at the fast lane.

ride a pedal bicycle afterwards (Phase 2). The first stage reported faster time of completion in the balance bike tests in cyclists than non-cyclists (very strong evidence for $H_1$) but no evidence for CMJ jump height. Thus, the balance bike manoeuvres demonstrated their capabilities to divide healthy adults with and without the ability to cycle. In the second stage, a meaningful balance bike training effect on balance bike performance (extreme evidence for

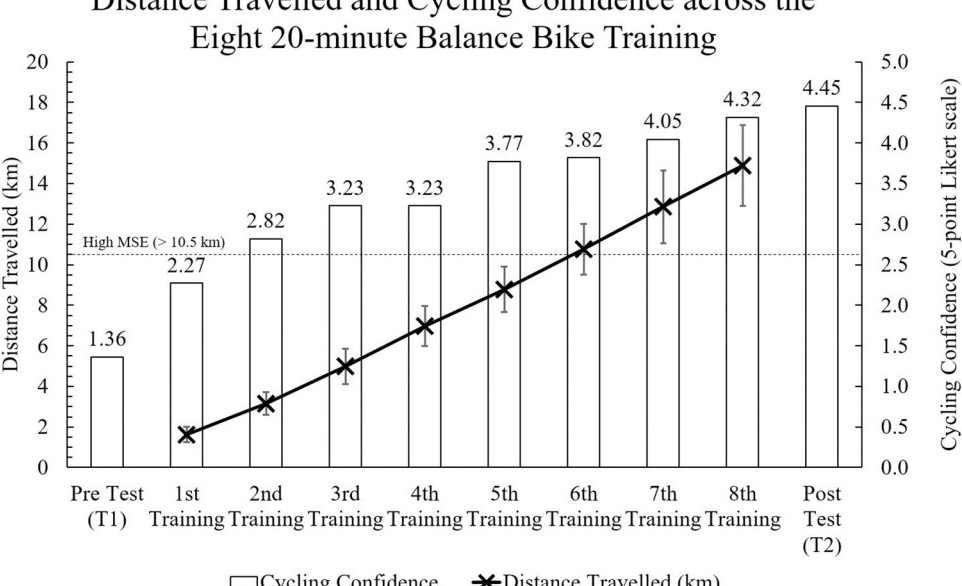

**Fig 4. Distance travelled and cycling confidence across the 8 × 20-minute balance bike training programme.** Note: MSE: Motor-skill engagement on a balance bike; Cycling confidence (1 = not confident; 2 = slightly confident; 3 = somewhat confident; 4 = fairly confident; 5 = very confident).

$H_1$) and the CMJ jump height (very strong evidence for $H_1$) were reported following the 8 × 20-minute balance bike training programme in the Phase 2. Noteworthily, all non-cyclists, with a wide range of age (19–63 y), learned to cycle a pedal bike independently after completing eight 20-minute sessions of our novel balance bike programme. They did not receive any instruction about pedalling a pedal bike in the sessions. This study provided evidence of positive riding skill transfer from a balance bike to a two-wheel bike and building confidence for every adult to learn how to cycle with dignity.

The balance bike manoeuvres designed by Kavanagh et al. [14] were adopted in this study. Previously, they assessed the ability of young children to ride a balance bike using the time of completion. In this study, we presented compelling evidence to support the efficacy of the two balance bike tests in distinguishing and assessing healthy adults with different cycling backgrounds on novel motor skills. We found that the balance bike testing results were 4.39 to 7.25 times more likely under the alternative hypothesis ($H_1$) than under the null hypothesis ($H_0$), yielding very strong evidence in favour of the alternative hypothesis. Our study revealed that cyclists had significantly faster completion times than non-cyclists on the 15 m sprint and the two-turn curved sprint, with times of 1.89 s and 3.31 s respectively. These findings support our hypotheses that cyclists would demonstrate superior performance in riding and controlling a balance bike when compared to non-cyclists.

The ability to maintain balance while cycling depends on various factors as cycling strategies, controlling lateral deviation, and level of expertise, which are important in restoring balance in unstable cycling situation [24]. Large differences in effect size have been observed in both straight and two-curved turn sprints, but lower limb muscle-related performance on the CMJ test is not found to be related to cycling capabilities. While a systematic review and meta-analysis have shown correlations between measures of balance and lower-extremity muscle strength/power in healthy individuals across the lifespan [25], it should be noted that the perturbed stance test, which was commonly used to measure balance in previous studies, may not

fully reflect the unique balance strategy and control required for cycling. Similarly, the CMJ test may not fully capture the motor coordination required for cycling. These findings support a previous study that suggests cycling is a standalone motor skill that derives from locomotion, object control or stability [14]. It is also important to note that the physical activity background of the participants in the study was not strictly controlled, meaning that there may have been participants with different sports experience in the non-cyclist group in Phase 1. Therefore, although the non-cyclists' CMJ performance improved in Phase 2, it was still lower than that of the non-cyclists in Phase 1 (28.42 ± 8.45 vs 35.13 ± 9.07 cm). The improvement in Phase 2 could be attributed to increased balance and/or lower limb muscle training (e.g., strengthening glutes, quadriceps, hamstrings, and calves) when riding a balance bike, which may have contributed to the increased CMJ height.

Concerning a bicycle's self-stability, continuing at a stable speed, and turning techniques play important roles. For typical cycling, 3.59 to 4.88 m·s$^{-1}$, predicted by the Whipple model, seems to be an optimal range to maintain in. Riding too fast or too slow would likely result in falling [26]. While the participants in this study only sprinted and glided on a balance bike in the balance bike 15 m sprint tests, in Phase 1, cyclists could still achieve an average of 3.24 m·s$^{-1}$ in the sprint test, while non-cyclists could only achieve 2.37 m·s$^{-1}$. It could be because cyclists tended to glide the balance bike, while non-cyclists tended to walk with the balance bike [5]. These patterns (i.e., walk, run, glide) have been previously categorized in a study where it was found that as riding speed on a balance bike increased, the frequency of foot contact with the ground decreased and locomotor modes with longer flight phases began to emerge [5]. Therefore, we decided that our balance bike training in phase 2 focused on teaching non-cyclists to attain a relatively high riding speed by gliding to attain bicycle self-stabilisation. In additional to riding speed, Cain et al. [27] reported that experienced cyclists tended to maintain cycling stability by trunk sway more than steering when compared with less experienced cyclists. A similar observation was made in the post-intervention interview where participants reported feeling more comfortable and capable of controlling the bike to glide and turn, and being able to relax their grip on the handlebars during riding after the training intervention. Altogether, practising gliding and leaning the body while turning at high speed would enhance cycling skills. The effectiveness of our Phase 2 balance bike training was evidenced by the shortened time of completion in the findings of Phase 2 by 1.68 s and 2.61 s in the two tracks, correspondingly. Participants might have acquired the ability to maintain balance on a balance bike and thus gain sufficient confidence to glide on it, as opposed to walking or running only.

Our original balance bike training programme had several strengths. We recognized that simply engaging in general training, such as strength training, is insufficient to improve balance function [28]. In order to maximize the transfer of learning and promote effective skill acquisition in real-world scenarios [8], a task-specified and multi-faceted approach is necessary. This approach was similar with Harper et al.'s perspective on learning in balance recovery intervention [29], which emphasizes training variety of exposing participants to a representative sample of postural disturbances and possible responses that they would encounter in daily life. Moreover, it highlights the need to provide training complexity to enhance cognitive involvement to aid in the generalization of learning. When learning to ride a balance bike, a task-specified approach can improve near transfer by focusing on specific skills required for balancing and control. However, a multi-faceted approach is needed for far transfer to apply these skills to real-world cycling scenarios. This includes navigating turns and obstacles on a cycling track, as well as maintaining control over their riding path when encountering other riders.

Kavanagh et al. [9] found that children (aged 4.5 ± 0.5 y) who rode their balance bike over greater distances (>10.5 km in 8 weeks) improved more on the time of completion in balance

bike tests. Our participants accumulated 14.88 ± 2.09 km, which is defined to have a high MSE [9]. Compared to previously published cycling training programmes that range from 135 [30] to 270 minutes [31], our study took a mid-range approach, with an accumulated training time of 160 minutes. This duration is relatively short, which is important since lack of time is a commonly reported barrier to engaging in structured exercise [32]. Accumulating multiple short bouts of exercise has been shown to provide similar health and fitness benefits as longer continuous exercise modes [33,34]. Additionally, short bouts of exercise provide more opportunities for individuals to fit exercise into a busy day. Our recent multiple-short bouts design allowed participants to have a time-efficient programme that could potentially maintain high motivation and reduce the accumulation of physical and mental fatigue and burden on the participants.

Furthermore, 100% of the participants in Phase 2 self-reported that their cycling confidence improved tremendously during the training programme and was able to cycle independently regardless of their age and cycling experience prior to our programme. This suggested that educating individuals to ride a balance bike greatly benefits learning to cycle independently. As noted in a previous systematic review [6], fear of falling is the biggest barrier to learning to cycle. However, our recent study found that most participants were able to reduce their fear of falling by regaining balance on the balance bike through the use of their feet and having greater control over their riding speed. This suggests that balance bike training can be effective method for reducing fear of falling not just for children and youth riders, but also for adult riders. Subsequently, to facilitate the natural balance response and allow it to fully develop, we followed the recommendation of Oddsson [28] and instructed the participants' instructors to refrain from providing "too early" external support, such as holding onto the handlebar, when the participants lost their balance on a balance bike. By doing so, we aimed to nurture and fully execute the natural balance responses of the participants. Our programme also included elements to disrupt cycling stability, e.g., stopping the balance bike at a rest zone and before stop signs and starting to ride, that built the sense of safe cycling on cycle lanes. There was no injury due to falling in our training. Positive cycling experience coupled with increased skill and confidence leads to more frequent cycling [16]. The balance bike and level ground allowed participants to feel safe so that their cycling confidence was gradually built across the sessions. As a result, participants learn to view the riding speed and fear of falling as controllable.

Real-world situations were introduced to learners concurrently after they master basic riding skills in our training [35–38]. The present programme incorporates a dual-task approach. This has been emphasised for assessing and training balance control in patients with balance impairments [39,40] and young adults [41]. By enhancing cognitive performance and thus cognitive activities during static and dynamic postures through dual-task training, falls could be prevented [40]. A nearly authentic environment, context and situation were created for task-based learning, i.e., a series of balance bike activities to challenge participants' riding manoeuvres and postural control, i.e., stability, in different scenarios were developed. Our training covered the essential skills that have been commonly identified [6] and the open skills demanded during cycling, i.e., reactions to the environment and other road users [42]. Participants were instructed to turn their heads when they were ready to turn. During head movements, they were instructed to look at specific spaces to determine the riding speed and timing of turning and even made precues loudly to further enhance attention on the traffic scenario. Indeed, the ability to handle multiple tasks is important to promote participants' cycling safety awareness. Thus, our training programme provided a platform for the participants to develop new cycling motor skills through balance bike experience in gaining confidence, balance, and speed control, and without neglecting the importance of understanding cycling safety applied on the cycling tracks. However, it is important to recognize the potential influence of cultural

differences on cycling behaviour. The recent design of the balance bike training programme focused on simulating cycling scenario on cycling tracks in Hong Kong, including the position of slow and fast lanes, cycling rules, and cycling etiquette. While this training was effective in improving participants' cycling confidence in these scenarios, future studies in other settings should consider necessary amendments to account for cultural differences and the unique challenges of cycling in different scenarios, like cycling on road. By doing so, participants may be better able to develop cycling confidence needed to navigate a variety of cycling environments.

Our study had several limitations that should be considered when interpreting the results. Firstly, the sample size was relatively small, which may limit the generalizability of the findings to other populations. Future studies with larger sample sizes would be beneficial to confirm the validity of the results and increase external validity. Additionally, the recruitment of adult participants was difficult, possibly due to concerns about their capability to learn to ride a balance bike and potential embarrassment about not being able to so. This may have affected the generalizability of the results. Future studies could explore ways to address these concerns and increase participation rates among adults, such as providing additional support and encouragement to participants or offering incentives to participate. Secondly, the study only included a one-group sample to test the intervention, which did not allow for a direct comparison to other training approach of learning to cycle, such as free play riding. A future study that includes a control group using different methods would be useful to compare the effectiveness of the balance bike intervention to other established methods.

Additionally, there were specific limitations related to our study design. Firstly, the task prioritisation of the participants was unknown, and future studies should investigate the effect of variable and fixed priority in dual task training. Several studies revealed additional advantage of variable priority over fixed priority [40,43,44]. Since our training did not explicitly prompt the participants to primarily focus on balance bike control or external stimuli, their task prioritisation strategy was unable to be determined. Further investigation is warranted. Secondly, the lower threshold of Motor Skill Engagement (MSE) for transferring stability skills from a balance bike to a two-wheel bike was not established in our study, as all non-cyclists in this study were able to cycle after the balance bike training. Some participants even accumulated over 10.5 km of riding distance (i.e., high MSE) after only six sessions. If a future study could recruit a large sample size with varied MSE and ability to gain cycling skills, more information could be discovered to shorten the training duration. Thirdly, the lack of strict control over the physical activity background of the participants in this study may have contributed to the heterogeneity of participants' backgrounds. Future studies should measure and control physical activity levels to include a more diverse population and make the findings more generalizable. Thirdly, previous study in children and youth have suggested that individual who engaged in physical activity less than twice a month tend to learn how to cycle later than those who engaged in physical activity on a daily basis. This highlights the importance of physical activity in developing cycling skills. However, in our study, the lack of strict control over the physical activity background of the participants may have contributed to the heterogeneity of participants' backgrounds. Future studies should measure and control physical activity levels to include a more diverse population and make the findings more generalizable.

## 5. Conclusion

This study provides valuable evidence for the effectiveness of balance bike tests and a novel balance bike intervention in teaching adults how to ride a bike. Specifically, our findings suggest that the novel balance bike training, which incorporated five principles (riding speed, gliding

to turn, cycling confidence, high MSE and a dual-task approach), was particularly effective in promoting balance bike performance and facilitating positive transfer of learning to cycling independently. The data attained from successful new cyclists could be further utilized to design a critical riding speed and minimum distance travel on a balance bike for optimal positive transfer of learning to ride a pedal bike. However, it is important to acknowledge the limitations of our study and the need for further research to confirm these findings and address these limitations. Nonetheless, this study provides valuable initial evidence and highlights the potential of the balance bike intervention as a useful tool for teaching cycling skills to adults.

## Supporting information

**S1 Appendix. STROBE checklist.**
(DOCX)

**S2 Appendix. Dataset for adult balance bike.**
(XLSX)

## Acknowledgments

We would like to express our sincere gratitude to our research assistant, Yu-Hin Kong and student-helpers for their invaluable assistance throughout the course of this project. Their dedication, diligence, and expertise have played a critical role in the success of this research endeavour. We are truly appreciative of their contributions to this manuscript.

## Author Contributions

**Conceptualization:** Gary C. C. Chow, Sophia C. W. Ha.

**Data curation:** Gary C. C. Chow.

**Formal analysis:** Gary C. C. Chow.

**Funding acquisition:** Gary C. C. Chow, Sophia C. W. Ha.

**Investigation:** Gary C. C. Chow, Sophia C. W. Ha.

**Methodology:** Gary C. C. Chow, Sophia C. W. Ha.

**Project administration:** Gary C. C. Chow, Sophia C. W. Ha.

**Resources:** Gary C. C. Chow.

**Software:** Sophia C. W. Ha.

**Supervision:** Gary C. C. Chow.

**Writing – original draft:** Gary C. C. Chow.

**Writing – review & editing:** Gary C. C. Chow, Sophia C. W. Ha.

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
