## [Decision Letter · Decision Letter 0]

5 Jun 2023

PONE-D-23-13359Positive Skill Transfer in Balance and Speed Control from Balance Bike to Pedal Bike in Adults: A Multiphase Intervention StudyPLOS ONE

Dear Dr. Chow,

Thank you for submitting your manuscript to PLOS ONE. After careful consideration, we feel that it has merit but does not fully meet PLOS ONE’s publication criteria as it currently stands. Therefore, we invite you to submit a revised version of the manuscript that addresses the points raised during the review process.

We look forward to receiving your revised manuscript.

Kind regards,

Bojan Masanovic, Ph.D.

Academic Editor

PLOS ONE

Journal Requirements:

"This work is supported by the Seed Funding Grant 2020/21 (RG 70/2020-2021R) from The Education University of Hong Kong. "

Reviewers' comments:

Reviewer's Responses to Questions

**Comments to the Author**

1. Is the manuscript technically sound, and do the data support the conclusions?

Reviewer #1: Partly

Reviewer #2: Partly

Reviewer #3: Partly

2. Has the statistical analysis been performed appropriately and rigorously? 

Reviewer #1: Yes

Reviewer #2: Yes

Reviewer #3: Yes

3. Have the authors made all data underlying the findings in their manuscript fully available?

Reviewer #1: Yes

Reviewer #2: Yes

Reviewer #3: Yes

4. Is the manuscript presented in an intelligible fashion and written in standard English?

Reviewer #1: No

Reviewer #2: Yes

Reviewer #3: Yes

5. Review Comments to the Author

Reviewer #1: The authors demonstrated the effectiveness of a balance bike intervention resulting in independent cyclists after the intervention. The study does appear to be novel, particularly because of the adult population. The entire manuscript should be reviewed for grammar and clarity. Please see more detailed comments below.

Introduction

Line 62-64, citations needed.

Opening paragraph seems to unnecessarily bounce between international and Hong Kong.

How do the 10 countries in the study of adults who do not know how to cycle relate to Hong Kong. In other words, what indicates that it is similar for Hong Kong.

Review entire document for English readability. Example: “These parents are difficult to coach their children to cycle which would lead to fewer individuals being able to cycle in a generation”

Line 75: explain the relevance of later cycling learning ages to the present study.

Line 76: how are these barriers relevant to current study – use of balance bike as an adult”

Line 100: I’m not sure that balance bike should be considered an emerging tool. They have been around for a long time, however they may be growing in popularity.

Line 105: oddly phrased: “To the best of our knowledge, it is plausible that adults may feel more comfortable”

Overall, the introduction can be more direct and linear leading to the purpose of the study. For example, COVID-19 is not the only or main reason for cycling popularity. It seems out of place to mention in the context of this study.

Additionally, you should include the studies that have been conducted using balance bikes. Here is just one for example: Mercê C, Branco M, Catela D, Lopes F, Cordovil R. Learning to Cycle: From Training Wheels to Balance Bike. International Journal of Environmental Research and Public Health. 2022; 19(3):1814. https://doi.org/10.3390/ijerph19031814

Methods

I would suggest providing all phase 1 details then phase 2 details rather than going back and forth for different phases of the methods.

Line 153: statement can be clarified “Step 2 was to record the times of completion of two anticipated balance bike manoeuvres.” Consider stating the “length of time to complete…” it is unclear what “anticipated” is referring to in this context.

Discussion

New information should not be presented in the discussion. For example, you discuss observations about cyclist v non-cyclist techniques of leaning while riding. There is no discussion of this in the results, were these observations systematically observed?

Similarly, you provide statements such as: The participants stated that they reduced their fear of falling as they could regain balance on the balance bike by putting their feet on the ground and could fully control their riding speed” which are not discussed as being collected systematically or in the results.

Line 48-50 Statement appears to be about your study, but has an external citation. Review for clarity.

Line 358 and subsequent paragraph: This seems to be describing more how you designed the intervention rather than a discussion of your findings. Consider moving this discussion to methods describing intervention development.

It seems that the limitation section can be expanded, for example more broadly issues with small sample size. You only used a one-group sample to test the intervention. You did not have a comparison to traditional methods (training wheels) to see if that could also produce independent cyclists after the same amount of training.

Reviewer #2: General Comments:

The manuscript addresses a very interesting and current topic, it presents an intervention program to learn to cycle in adults, which also brings innovation.

However, the methodological component has several weaknesses, it is necessary to explain in more detail the process of developing the program, supporting it with an adequate theoretical and conceptual framework. The sample is very small so the conclusions are, in some cases, abusive. It is necessary to clarify the limitations of the study.

Introduction

Lines 81-83 – The balance bike affords several and different types of locomotion pattern besides “walk” and “sprint”. Even this variety of patterns is pointed out as one of the causes for their greater efficiency during learning. This is a point worth exploring further in the discussion. Suggested reading: doi.org/10.3390/children9121937

Line 84 – please include a reference.

Line 89 - The reference used only gives suggestions for using the balance bike as preparation for traditional cycling. However, this transition or transfer is not studied by these authors. You should include references that study this transfer. Suggested reading: doi/full/10.1080/17408989.2021.2005014

Line 93 - The reference used does not include adult participants, which is the target population of the present manuscript. In this sense, authors should seek to add references with their target population if possible. And, clarify the Kavanagh’s study population.

Lines 98-104 - There are already several studies that support why the balance bike enables this balance exploration and training, and consequently promotes a more effective and efficient transfer to traditional cycling. Please use references and avoid "Anecdotal evidences", the suggested readings in the comments above may help.

Methods

Lines 137-139 - Please explain the reasons for the definition of the criterion cyclist/non-cyclist. The ability to brake safely was not considered, why?

Line 144 – Sample - The sample size is quite small. This is a weakness that may compromise the external validity of the results. This limitation should be discussed and clearly presented in the article's discussion.

Lines 163-178 - It is mentioned in the introduction that one of the objectives is to develop an intervention programme with the balance bike for adults. However, this presentation of the programme only identifies a guiding principle (dual-task). How did this development process take place? Who developed it, a motor learning specialist? It is also mentioned that the observational study (phase 1) supports the development of the programme, how? Can you give some practical examples?

Lines 173, 175-176 - Please clarify the speed condition. Initially you state that the speed was self-selected by the participant, subsequently you state that you were asked to run the course at the highest possible speed. The speed is a variable that influences the learning with the balance bike, this should be a variable to explore in the programme.

Table 1 - Table 1 mentions various cycling patterns with the balance bike such as "walk" "run" and "glide". These patterns have already been categorized in a previous study, you should reference the study and clearly present what you mean by each of the patterns you are referencing.

Table 1 - From the description of the exercises it is possible to see that the manipulation of speed was taken into account. Please explain in the programme presentation the role of speed in the learning of cycling and how you manipulated it in order to promote this learning.

Lines 201-202 – Please introduce a reference.

Line 208 - Please clarify if this scale has been validated/used in any other previous studies or if it was developed for the present manuscript.

Lines 217-223 - Please clarify "positive transfer" by supporting it with references. What criteria are used for the distinction between "quick positive motor skill transfer" "late positive motor skill transfer" and "zero transfer"? These criteria for distinction should be clarified and supported by references.

Discussion

CMJ – There is a concern for the application of the CMJ test, being that it is then compared and reported between groups. However, the reason for the "importance" or connection of this test with the balance bike is not clear. Please clarify it.

293-295 – Please clarify this sentence, I was not able to understand it meaning.

297 - It mentions "one of our 4 hypotheses", I cannot find the clear presentation of the hypotheses in the text.

331 - They address the strengths of the intervention programme presented. What are its weaknesses? Please identify above (in the programme description) what the participant/instructor ratio is.

333 – “Training should be a task-specified and multi-faceted approach.” I personally fully agree with this statement. But it should not be presented without contextualisation, that is, what is the theoretical or conceptual basis that supports it? This should be the same as that which supports the development of the programme and which, until now, has been omitted.

333-334 - “Adopting a balance bike in our training provided cycling-specific lateral deviations to the riders.” How can the authors state this so surely? Have the lateral deviations been measured?

346-348 - The fear of falling is identified in literature as one of the biggest barriers to learning to cycle on a traditional bicycle, strengthen this sentence with literature references.

358 – “Real-world situations”? Did the programme include a component of cycling on the road with car traffic?

385 – Limitations - Physical activity influences learning to cycle in children (there is at least one recent study that finds this association), can you strengthen this presentation with references. Another major limitation was not controlling for participants' prior experience on various bicycles. We do not know if the previous experience influenced the type of transfer found "quick positive motor skill transfer" "late positive motor skill transfer" and "zero transfer", however, considering that learning can be seen as the sum of experiences, we can assume that yes. This weakness should be presented and discussed.

392 – “Extreme evidence” based on a study of only 11 adults is abusive. please reconsider the use of this expression.

393 – How did you arrive at these 5 principles? This is a discussion to complement the discussion section.

Reviewer #3: This manuscript proposes to study the transfer of skills between balance cycling and pedaling in adults.

The authors are to be congratulated for the methodological rigor they try to imprint in this study, however some questions arise as you read:

- Both the title and the second objective refer to transfer between motor skills, however, the authors do not describe what they mean by positive transfer or what is meant by transfer in general. It is recommended that this be done in the framework.

- Another aspect is that it is not made clear, which skill is meant to assess transfer. The authors should be clear in this definition, since the act of pedaling is not present in the first bicycle. This should be clarified.

In the Methods section - The bicycle being the means to test the transfer, the difference between the bicycles used for each condition is not explained.

- In the Methods section, ln. 117-118 - the characteristics of each phase are presented. In phase 2, what is the rationale for choosing 8x20-minutes? Why not more or less time or sessions?

- In the Methods section, ln. 142 - anthropometric measurements were performed. What measurement protocols were used? What expertise does the measurer have?

- In the Methods section, ln. 146 - the cultural context in motor learning is very important, as well as the learning possibilities between sexes, decades of birth and their motor availability. This difference between elements of each sex is a weakness of the study. To bridge these differences, assessing motor competence would be recommended. Did you do this? How do you support the results in light of the contexts?

- In the Methods section, ln. 166 - "Challenge-Taking Course." A schematic/image illustrating the course would be welcomed. This allows the principle of replicability to be guaranteed.

- In the Methods section, ln. 172 - "Solve real-world challenges": Which ones? How were they simulated in this situation?

- In the Methods section, ln. 201-202 - There are several ways to calculate the optimal saddle height. The way they use is unclear, especially the "slightly bent" of the knees and hips. Can they be clearer? Have you not used any protocols?

- In the Methods section, ln. 219 - Is the classification "Quick positive transfer" or "late positive transfer" your classification or used by other authors? What is the rationale for having 15 minutes as the cut-off value? How does a person who can perform the task in 15 minutes differ so much from one who can do it in 16 minutes?

- In the Results section, ln. 266 - The authors indicate that confidence in cycling was assessed across sessions. What results come from this...is there a session that shows a jump in confidence in bicycling? I recommend statistical treatment of these data.

- In the Results section, ln. 303 - Was stability assessed in any way across sessions? No reference to such an evaluation appears, which makes the use of this term abusive.

6. PLOS authors have the option to publish the peer review history of their article (what does this mean?). If published, this will include your full peer review and any attached files.

Reviewer #1: No

Reviewer #2: No

Reviewer #3: **Yes: **Marco Branco

---

## [Author Response · Author response to Decision Letter 0]

3 Jul 2023

Journal: PLOS ONE Decision: Revision required [PONE-D-23-13359]

Manuscript Status: Pending major revisions

Manuscript ID: PONE-D-23-13359

Type: Article

Positive Skill Transfer in Balance and Speed Control from Balance Bike to Pedal Bike in Adults: A Multiphase Intervention Study

Point-by-Point Response

Reviewer 1

Point raised by Reviewer #1: C01

The authors demonstrated the effectiveness of a balance bike intervention resulting in independent cyclists after the intervention. The study does appear to be novel, particularly because of the adult population. The entire manuscript should be reviewed for grammar and clarity. Please see more detailed comments below.

Response by author: 

Thanks for the recognition and we are going to revise the manuscript to address the suggestions.

Point raised by Reviewer #1: C02

Introduction

Line 62-64, citations needed.

Response by author:

Those sentences shared the same citation of

Research Office Legislative Council Secretariat. Bicycle-friendly policy in Hong Kong. Transport. February 4, 2021. Accessed 28 Aug, 2022. https://www.legco.gov.hk/research-publications/english/2021issh17-bicycle-friendly-policy-in-hong-kong-20210204-e.pdf

Therefore, the introduction has been revised “According to Research Office Legislative Council Secretariat [3], the total length of cycle tracks in Hong Kong increased from 206.8 km in late 2011 to 225.2 km in early 2020, representing a nearly 9% increase. In addition, the government plans to build an additional 104.6 km of cycle tracks in the foreseeable future. Despite Hong Kong is also seeking a move to become more bicycle-friendly like other regions [2] by building and extending cycle paths, Hong Kong had a low bicycle usage (i.e., percentage of people using bicycles in everyday life in each city) of 0.5% compared with its regional counterparts [3].”

Point raised by Reviewer #1: C03

Opening paragraph seems to unnecessarily bounce between international and Hong Kong.

Response by author 

Accepted the comments and revised the opening paragraph.

Line 56 – 71, Para 1.

Point raised by Reviewer #1: C04

How do the 10 countries in the study of adults who do not know how to cycle relate to Hong Kong. In other words, what indicates that it is similar for Hong Kong.

Response by author 

Accepted the comments and revised the sentences to make it clear.

“However, recent data noticed that an average of 9.7% of the total population in 10 countries do not know how to cycle [2]. According to Research Office Legislative Council Secretariat [3], the total length of cycle tracks in Hong Kong increased from 206.8 km in late 2011 to 225.2 km in early 2020, representing a nearly 9% increase. In addition, the government plans to build an additional 104.6 km of cycle tracks in the foreseeable future. Despite Hong Kong is also seeking a move to become more bicycle-friendly like other regions [2] by building and extending cycle paths, Hong Kong had a low bicycle usage (i.e., percentage of people using bicycles in everyday life in each city) of 0.5% compared with its regional counterparts [3].”

Point raised by Reviewer #1: C05

Review entire document for English readability. Example: “These parents are difficult to coach their children to cycle which would lead to fewer individuals being able to cycle in a generation”

Response by author 

The entire document has been revised for English readability. 

E.g., Line 69 – 84 

“Learning to cycle independently is a common rite of passage for children, often taught by their parents as they approach school age. In recent years, the average age for achieving this milestone has decreased. Children born between 2000–2019 now learn to cycle independently at an average age of 5.22 y, compared to 5.68 y for those born between 1980–1999 and 6.21 y for those born between 1960–1979 [2]. This trend can be attributed to the increasing availability of dedicated cycling tracks and child-size-friendly bicycles, which make it easier for children to learn at an earlier age. However, some parents may face challenges teaching their children to cycle due to concerns about their own cycling safety [4]. A recent cycling study in Hong Kong found that 15.6% of parents do not know how to ride a two-wheel bike [5], which can limit their ability to teach their children to cycle. For many individuals, not being able to cycle independently as an adult can often be tracked back to ineffective learning approaches [6] or insufficient childhood playtime with parents [2]. Negative learning experiences, such as falling while learning on a two-wheel bike, can discourage them from continuing to learn. Similarly, using a bicycle with training wheels can also be ineffective in acquiring essential cycling skills and stability [7], and may delays the age at which individuals learn to cycle independently [6]. However, in recent years, balance bikes have become a popular alternative to overcome these challenges. These bikes lack pedals and allows rider to walk or sprint while seated [8], making it easier to acquire the necessary skills and improve cycling stability. In fact, studies have shown that a balance bike can be more effective in helping riders learn the essential skill of maintaining balance on two wheels, compared to bikes with training wheels that focus on pedalling skills [6, 7]. With the right support, including access to affordable, appropriately sized adult balance bikes and effective learning design, individuals of all ages can overcome these barriers and learn to cycle independently.”

Point raised by Reviewer #1: C06

Line 75: explain the relevance of later cycling learning ages to the present study.

Response by author 

We revise the sentences to make it clear.

“…This trend can be attributed to the increasing availability of dedicated cycling tracks and child-size-friendly bicycles, which make it easier for children to learn at an earlier age. However, some parents may face challenges teaching their children to cycle due to concerns about their own cycling safety [4]. A recent cycling study in Hong Kong found that 15.6% of parents do not know how to ride a two-wheel bike [5], which can limit their ability to teach their children to cycle.”

Point raised by Reviewer #1: C07

Line 76: how are these barriers relevant to current study – use of balance bike as an adult”

Response by author 

The paragraph has been revised.

“…However, in recent years, balance bikes have become a popular alternative to overcome these challenges. These bikes lack pedals and allows rider to walk or sprint while seated [8], making it easier to acquire the necessary skills and improve cycling stability. In fact, studies have shown that a balance bike can be more effective in helping riders learn the essential skill of maintaining balance on two wheels, compared to bikes with training wheels that focus on pedalling skills [6, 7]. With the right support, including access to affordable, appropriately sized adult balance bikes and effective learning design, individuals of all ages can overcome these barriers and learn to cycle independently.”

Point raised by Reviewer #1: C08

Line 100: I’m not sure that balance bike should be considered an emerging tool. They have been around for a long time, however they may be growing in popularity.

Response by author 

We agree the observation and has been revised the sentences.

“…However, in recent years, balance bikes have become a popular alternative to overcome these challenges. These bikes lack pedals and allows rider to walk or sprint while seated [8], making it easier to acquire the necessary skills and improve cycling stability. In fact, studies have shown that a balance bike can be more effective in helping riders learn the essential skill of maintaining balance on two wheels, compared to bikes with training wheels that focus on pedalling skills [6, 7]. With the right support, including access to affordable, appropriately sized adult balance bikes and effective learning design, individuals of all ages can overcome these barriers and learn to cycle independently.”

Point raised by Reviewer #1: C09

Line 105: oddly phrased: “To the best of our knowledge, it is plausible that adults may feel more comfortable”

Response by author 

We revised the sentence as

“In fact, studies have shown that a balance bike can be more effective in helping riders learn the essential skill of maintaining balance on two wheels, compared to bikes with training wheels that focus on pedalling skills [6, 7]. With the right support, including access to affordable, appropriately sized adult balance bikes and effective learning design, individuals of all ages can overcome these barriers and learn to cycle independently.”

Point raised by Reviewer #1: C10

Overall, the introduction can be more direct and linear leading to the purpose of the study. For example, COVID-19 is not the only or main reason for cycling popularity. It seems out of place to mention in the context of this study.

Additionally, you should include the studies that have been conducted using balance bikes. Here is just one for example: Mercê C, Branco M, Catela D, Lopes F, Cordovil R. Learning to Cycle: From Training Wheels to Balance Bike. International Journal of Environmental Research and Public Health. 2022; 19(3):1814. https://doi.org/10.3390/ijerph19031814

Response by author 

Thanks for the comments and the introduction has been revised. 

Additional references are also cited in the recent study.

Point raised by Reviewer #1: C11

Methods

I would suggest providing all phase 1 details then phase 2 details rather than going back and forth for different phases of the methods.

Response by author 

Thanks for the comments. We consider the suggestions deeply and made some rearrangements to present the study clearly.

Point raised by Reviewer #1: C12

Line 153: statement can be clarified “Step 2 was to record the times of completion of two anticipated balance bike manoeuvres.” Consider stating the “length of time to complete…” it is unclear what “anticipated” is referring to in this context.

Response by author 

Sentences were revised accordingly,

“In Phase 1, the participants’ demographics (i.e., age and gender), anthropometrics (i.e., height and weight), and physical functional capacity (i.e., jump height of countermovement jump [CMJ]) were measured. Following that, the times of completion for two assigned balance bike manoeuvres were recorded.”

Point raised by Reviewer #1: C13

Discussion

New information should not be presented in the discussion. For example, you discuss observations about cyclist v non-cyclist techniques of leaning while riding. There is no discussion of this in the results, were these observations systematically observed?

Response by author

Thanks for the comments. We tried to revise the discussion in our study to make it clear with evidence support.

“…A similar observation was made in the post-intervention interview where participants reported feeling more comfortable and capable of controlling the bike to glide and turn, and being able to relax their grip on the handlebars during riding after the training intervention.”

Point raised by Reviewer #1: C14

Similarly, you provide statements such as: The participants stated that they reduced their fear of falling as they could regain balance on the balance bike by putting their feet on the ground and could fully control their riding speed” which are not discussed as being collected systematically or in the results.

Response by author 

At T2, researcher interviewed the participants and video recorded their responses. To clarify this missing procedure, a new section was added in the methods and the results.

“Methods

… At T2, after undergoing an identical physical functional capacity test, participants were verbally interviewed about their perception of the learning experience. They were invited to self-launch and ride a pedal bike to determine whether they could be up-skilled as a cyclist after the balance bike training.”

“Self-perception of the learning experience

Before the pedal bike trial at T2, a structured interview was conducted to assess the participants’ perception regarding their acquisition of cycling skills during the programme. The interview lasted for 3 minutes and consisted of two questions: Q1) How did you feel after completing the balance bike training programme? and Q2) Which specific elements do you feel you developed after completing the balance bike training programme that contributed to changes of your cycling confidence?”

“Results

Based on the verbal survey conducted during Phase 2 of the study, participants reported feeling fairly to very high level of confidence with their cycling abilities, with a score of 4.45 ± 0.52 out of possible 5, after completing all training sessions. Participants commented that after completing the balance training programme (Q1), they felt comfortable riding a balance bike (n = 11), experienced a reduction in their fear of falling (n = 6), and felt more relaxed while holding onto the handles of the bike (n = 4). Specifically, in response to Q2, participants commented on the improvements in their cycling confidence, including an increased capacity of controlling the bike during gliding and turning (n = 7), and an increase in their skills and knowledge for handling the traffic chaos (n = 4)…”

Point raised by Reviewer #1: C15

Line (3)48-50 Statement appears to be about your study, but has an external citation. Review for clarity.

Response by author 

We revised the sentences to clarify the message.

“Subsequently, to facilitate the natural balance response and allow it to fully develop, we followed the recommendation of Oddsson [30] and instructed the participants’ instructors to refrain from providing “too early” external support, such as holding onto the handlebar, when the participants lost their balance on a balance bike. By doing so, we aimed to nurture and fully execute the natural balance responses of the participants.”

Point raised by Reviewer #1: C16

Line 358 and subsequent paragraph: This seems to be describing more how you designed the intervention rather than a discussion of your findings. Consider moving this discussion to methods describing intervention development.

Response by author 

We consider the suggestion deeply, but we feel that it is important to include a discussion of the design of intervention (i.e., balance bike training programme) in the discussion section.

Some amendment was made in this paragraph to highlight the recommendation to the future studies.

“…However, it is important to recognize the potential influence of cultural differences on cycling behaviour. The recent design of the balance bike training programme focused on simulating cycling scenario on cycling tracks in Hong Kong, including the position of slow and fast lanes, cycling rules, and cycling etiquette. While this training was effective in improving participants’ cycling confidence in these scenarios, future studies in other settings should consider necessary amendments to account for cultural differences and the unique challenges of cycling in different scenarios, like cycling on road. By doing so, participants may be better able to develop cycling confidence needed to navigate a variety of cycling environments.”

Point raised by Reviewer #1: C17

It seems that the limitation section can be expanded, for example more broadly issues with small sample size. You only used a one-group sample to test the intervention. You did not have a comparison to traditional methods (training wheels) to see if that could also produce independent cyclists after the same amount of training.

Response by author 

We agreed that our study had several limitations, so we added this concern into the discussion. However, consider we have sufficient evidence to know traditional methods may negatively affect the learning of cycle, we made the sentences as,

“...Secondly, the study only included a one-group sample to test the intervention, which did not allow for a direct comparison to other training approach of learning to cycle, such as free play riding. A future study that includes a control group using different methods would be useful to compare the effectiveness of the balance bike intervention to other established methods.”

Reviewer #2

Point raised by Reviewer #2 C01

General Comments:

The manuscript addresses a very interesting and current topic, it presents an intervention program to learn to cycle in adults, which also brings innovation.

However, the methodological component has several weaknesses, it is necessary to explain in more detail the process of developing the program, supporting it with an adequate theoretical and conceptual framework. The sample is very small so the conclusions are, in some cases, abusive. It is necessary to clarify the limitations of the study.

Response by author 

Thanks for the comments. We revise the manuscript to address the suggestions.

Point raised by Reviewer #2 C02

Introduction

Lines 81-83 – The balance bike affords several and different types of locomotion pattern besides “walk” and “sprint”. Even this variety of patterns is pointed out as one of the causes for their greater efficiency during learning. This is a point worth exploring further in the discussion. Suggested reading: doi.org/10.3390/children9121937

Response by author 

Sentences in Introduction section has been revised.

“These bikes lack pedals and allows rider to walk or sprint while seated [8], making it easier to acquire the necessary skills and improve cycling stability.”

In the discussion, we added,

“…It could be because cyclists tended to glide the balance bike, while non-cyclists tended to walk with the balance bike [8]. These patterns (i.e. walk, run, glide) have been previously categorized in a study where it was found that as riding speed on a balance bike increased, the frequency of foot contact with the ground decreased and locomotor modes with longer flight phases began to emerge [8].”

Point raised by Reviewer #2 C03

Line 84 – please include a reference.

Response by author 

References had been added.

Point raised by Reviewer #2 C04

Line 89 - The reference used only gives suggestions for using the balance bike as preparation for traditional cycling. However, this transition or transfer is not studied by these authors. You should include references that study this transfer. Suggested reading: doi/full/10.1080/17408989.2021.2005014

Response by author 

Agree the comment. Reference had been replaced.

“Riding on a balance bike has been proven to be an effective method for positively transferring the skills needed to learn how to cycle [9].”

Point raised by Reviewer #2 C05

Line 93 - The reference used does not include adult participants, which is the target population of the present manuscript. In this sense, authors should seek to add references with their target population if possible. And, clarify the Kavanagh’s study population.

Response by author 

To clarify the meaning, the sentences had been revised.

“Learning to control a balance bike is attainable, and even preschool children with varying fundamental motor skill level could improve the capacity of balance bike control with eight weeks of free play on a balance bike [12]. Moreover, a higher volume of practice, known as motor skill engagement (MSE), on a balance bike led to more improvement in balance bike skills [12].”

Point raised by Reviewer #2 C06

Lines 98-104 - There are already several studies that support why the balance bike enables this balance exploration and training, and consequently promotes a more effective and efficient transfer to traditional cycling. Please use references and avoid "Anecdotal evidences", the suggested readings in the comments above may help.

Response by author 

Thanks for the comments. We had revised the sentence to refine the message.

“In fact, studies have shown that a balance bike can be more effective in helping riders learn the essential skill of maintaining balance on two wheels, compared to bikes with training wheels that focus on pedalling skills [6, 7].”

Point raised by Reviewer #2 C07

Methods

Lines 137-139 - Please explain the reasons for the definition of the criterion cyclist/non-cyclist. The ability to brake safely was not considered, why?

Response by author 

To clarify the definition, a new section was added before Participant section,

“Study design

 In this study, participants were classified into two groups: cyclist group and non-cyclist group. Studies typically identified three criteria for independent cycling: the ability to self-launch, riding a bike continuously for a certain distance, and braking without assistance [9]. However, the combination of criteria varied across studies. Since braking using the handlebar is not part of a balance bike’s design, participants in this study who were able to self-launch and ride a pedal bike for 5 minutes continuously were classified as the cyclist group, while those who were unable to do that were classified as the non-cyclist group.”

Point raised by Reviewer #2 C08

Line 144 – Sample - The sample size is quite small. This is a weakness that may compromise the external validity of the results. This limitation should be discussed and clearly presented in the article's discussion.

Response by author 

We agreed that our study had several limitations, so we added this concern into the discussion.

“…Firstly, the sample size was relatively small, which may limit the generalizability of the findings to other populations. Future studies with larger sample sizes would be beneficial to confirm the validity of the results and increase external validity. Additionally, the recruitment of adult participants was difficult, possibly due to concerns about their capability to learn to ride a balance bike and potential embarrassment about not being able to so. This may have affected the generalizability of the results. Future studies could explore ways to address these concerns and increase participation rates among adults, such as providing additional support and encouragement to participants or offering incentives to participate.”

Point raised by Reviewer #2 C09

Lines 163-178 - It is mentioned in the 

introduction that one of the objectives is to develop an intervention programme with the balance bike for adults. 

However, this presentation of the programme only identifies a guiding principle (dual-task). How did this development process take place? Who developed it, a motor learning specialist? It is also mentioned that the observational study (phase 1) supports the development of the programme, how? Can you give some practical examples?

Response by author 

We clarified the message in two sections

 “Methods

Balance bike training programme development

In Phase 1, key cycling performance variables, issues, and preferences were determined by combing participants’ functional capacity, balance bike testing data, perceived confidence in two groups. Other factors, such as cycle patterns on balance bike, previous training experience, appropriate training features and designs, and offering autonomy support in the training, were also considered in the development of balance bike training programme. After completing data collection in Phase 1, a sport scientist developed the balance bike training programme to address the needs of non-cyclists.”

“Results

Phase 1: Balance bike training programme development

The balance bike programme was based on five principles: riding speed, gliding to turn, cycling confidence, high MSE, and a dual-task approach. Each session was tailored to participants’ riding abilities, and an instructor-to-participant ratio of 1: 2 to 1: 4 was used regarding the participants’ availability. The training approach used an authentic learning approach that presented challenges resembling real-world scenarios. Table 1 provides an overview of the balance bike intervention framework…

… The programme was designed as a series of challenges that simulated different scenarios encountered on cycling tracks, such as navigating traffic chaos or taking turns leading the group, to encourage participants to adapt to unexpected situations for better cycling stability.”

Point raised by Reviewer #2 C10

Lines 173, 175-176 - Please clarify the speed condition. Initially you state that the speed was self-selected by the participant, subsequently you state that you were asked to run the course at the highest possible speed. The speed is a variable that influences the learning with the balance bike, this should be a variable to explore in the programme.

Response by author 

We clarified the message in the 

“Results

Phase 1: Balance bike training programme development …

Given faster times of completion of 15m sprint and two-turn curved sprint in cyclist, the programme emphasized riding speed and gliding to turn throughout the programme. Participants were encouraged to practice the route as much as possible during the standardized 20-minute session to achieve a longer travelling distance or more successful completion of task. At the end of each session, the instructor shared the distance travelled by each participant using cycle computer, allowing participants to track their progress over time. 

Due to the previous poor cycling experience of adult non-cyclist, autonomy support and mastery of the skills were the key variable for their active participation (i.e., MSE) with confidence, participants had the freedom to choose the speed, skill, and level of participation, including when to stop at rest area or use the overtaking lane. The instructor encouraged participants to ride as fast as they could, but also provided acceptance to participants who preferred to ride at their own pace, even if it was slower, with the aim of ensuring that participants felt comfortable and in control of the bike while also pushing themselves to improve their performance.”

Point raised by Reviewer #2 C11

Table 1 - Table 1 mentions various cycling patterns with the balance bike such as "walk" "run" and "glide". These patterns have already been categorized in a previous study, you should reference the study and clearly present what you mean by each of the patterns you are referencing.

Response by author 

We support the suggestions and revised the sentences in the discussion section.

“It could be because cyclists tended to glide the balance bike, while non-cyclists tended to walk with the balance bike [8]. These patterns (i.e. walk, run, glide) have been previously categorized in a study where it was found that as riding speed on a balance bike increased, the frequency of foot contact with the ground decreased and locomotor modes with longer flight phases began to emerge [8].”

Point raised by Reviewer #2 C12

Table 1 - From the description of the exercises it is possible to see that the manipulation of speed was taken into account. Please explain in the programme presentation the role of speed in the learning of cycling and how you manipulated it in order to promote this learning.

Response by author 

Riding speed was taken into account and emphasized a high priority during the training session. We revised the sentences in “Motor-skill engagement (MSE) on a balance bike” to highlight this important message.

“Results

Phase 1: Balance bike training programme development …

Given faster times of completion of 15m sprint and two-turn curved sprint in cyclist, the programme emphasized riding speed and gliding to turn throughout the programme. Participants were encouraged to practice the route as much as possible during the standardized 20-minute session to achieve a longer travelling distance or more successful completion of task. At the end of each session, the instructor shared the distance travelled by each participant using cycle computer, allowing participants to track their progress over time.”

Point raised by Reviewer #2 C13

Lines 201-202 – Please introduce a reference.

Response by author 

The sentences had been revised and two references were cited.

“The seat height of the bike was then adjusted to fit each participant’s body, enabling them to place both feet on the ground with slightly bent hips and knees [12, 17]. This position allowed for proper extension of the hip and knee joints when propelling themselves forward.”

Point raised by Reviewer #2 C14

Line 208 - Please clarify if this scale has been validated/used in any other previous studies or if it was developed for the present manuscript.

Response by author 

We have revised the section.

“Previous studies have used self-report cycling confidence to examine the effectiveness of cycling training among adults [19, 20]. In this study, cycling confidence was assessed verbally after each of the eight training sessions and the two testing sessions (i.e., T1 & T2), resulting in a total of 10 recorded confidence scores. Participants were asked to rate their confidence to ride a pedal bike on a five-point Likert scale (1 = not confident; 2 = slightly confident; 3 = somewhat confident; 4 = fairly confident; 5 = very confident).”

Point raised by Reviewer #2 C15

Lines 217-223 - Please clarify "positive transfer" by supporting it with references. 

What criteria are used for the distinction between "quick positive motor skill transfer" "late positive motor skill transfer" and "zero transfer"? These criteria for distinction should be clarified and supported by references.

Response by author 

Our original thought is to use a self-designed time frame to define “the speed of learning in transfer of learning”. However, we understand that this definition does not have sufficient evidence to support, therefore, we decided to report the time intervals of successful transfer only.

“Cycle Independently

… The time taken by participants to cycle independently was recorded and reported, in 15-minute intervals: below 15 minutes, 16 to 30 minutes, 30 to 45 minutes, and 45 to 60 minutes. The trial had a maximum time limit of 60 minutes.”

“Results

Phase 2: Assessment for balance bike Intervention

… Out of the eleven participants, eight were able to self-launch and ride a pedal bike continuously within 15 minutes, while the remaining three were able to cycle independently within 16 to 30 minutes (n = 1) and 45 to 60 minutes (n = 2).”

Point raised by Reviewer #2 C16

Discussion

CMJ – There is a concern for the application of the CMJ test, being that it is then compared and reported between groups. However, the reason for the "importance" or connection of this test with the balance bike is not clear. Please clarify it.

Response by author 

We try to clarify the reason in the discussion

“…While a systematic review and meta-analysis have shown correlations between measures of balance and lower-extremity muscle strength/power in healthy individuals across the lifespan…”

Point raised by Reviewer #2 C17

293-295 – Please clarify this sentence, I was not able to understand it meaning.

Response by author 

We revised the sentences to explain how we use the Bayes factor to indicate the efficacy of the selected balance bike test on distinguishing cyclist and non-cyclist in adults.

“In this study, we presented compelling evidence to support the efficacy of the two balance bike tests in distinguishing and assessing healthy adults with different cycling backgrounds on novel motor skills. We found that the balance bike testing results were 4.39 to 7.25 times more likely under the alternative hypothesis (H1) than under the null hypothesis (H0), yielding very strong evidence in favor of the alternative hypothesis.”

Point raised by Reviewer #2 C18

297 - It mentions "one of our 4 hypotheses", I cannot find the clear presentation of the hypotheses in the text.

Response by author 

Thanks for the comments. We revised the statement to make it clear.

“Our study revealed that cyclists had significantly faster completion times than non-cyclists on the 15 m sprint and the two-turn curved sprint, with times of 1.89 s and 3.31 s respectively. These findings support our hypotheses that cyclists would demonstrate superior performance in riding and controlling a balance bike when compared to non-cyclists.”

Point raised by Reviewer #2 C19

331 - They address the strengths of the 

intervention programme presented. What are its weaknesses? 

Please identify above (in the programme description) what the participant/instructor ratio is.

We made some amendments in the discussion.

“…However, it is important to recognize the potential influence of cultural differences on cycling behaviour. The recent design of the balance bike training programme focused on simulating cycling scenario on cycling tracks in Hong Kong, including the position of slow and fast lanes, cycling rules, and cycling etiquette. While this training was effective in improving participants’ cycling confidence in these scenarios, future studies in other settings should consider necessary amendments to account for cultural differences and the unique challenges of cycling in different scenarios, like cycling on road. By doing so, participants may be better able to develop cycling confidence needed to navigate a variety of cycling environments.”

To clarify the participant/instructor ratio, we made clarification in the results section.

“Results

Phase 1: Balance bike training programme development

The balance bike programme was based on five principles: riding speed, gliding to turn, cycling confidence, high MSE, and a dual-task approach. Each session was tailored to participants’ riding abilities, and an instructor-to-participant ratio of 1: 2 to 1: 4 was used regarding the participants’ availability.”

Point raised by Reviewer #2 C20

333 – “Training should be a task-specified and multi-faceted approach.” 

I personally fully agree with this statement. But it should not be presented without contextualisation, that is, what is the theoretical or conceptual basis that supports it? This should be the same as that which supports the development of the programme and which, until now, has been omitted.

Response by author 

Citation is added and sentences were revised to clarify the message.

“Our original balance bike training programme had several strengths. We recognized that simply engaging in general training, such as strength training, is insufficient to improve balance function [30]. In order to maximize the transfer of learning and promote effective skill acquisition in real-world scenarios [11], a task-specified and multi-faceted approach is necessary. This approach was similar with Harper et al.’s perspective on learning in balance recovery intervention .... However, a multi-faceted approach is needed for far transfer to apply these skills to real-world cycling scenarios. This includes navigating turns and obstacles on a cycling track, as well as maintaining control over their riding path when encountering other riders.”

Point raised by Reviewer #2 C21

333-334 - “Adopting a balance bike in our training provided cycling-specific lateral deviations to the riders.” How can the authors state this so surely? Have the lateral deviations been measured?

Response by author 

In recent study, we did not measure the lateral deviation, so we revised the sentences to make it clear and concise. 

“Our original balance bike training programme had several strengths. We recognized that simply engaging in general training, such as strength training, is insufficient to improve balance function [30]. In order to maximize the transfer of learning and promote effective skill acquisition in real-world scenarios [11], a task-specified and multi-faceted approach is necessary. This approach was similar with Harper et al.’s perspective on learning in balance recovery intervention .... However, a multi-faceted approach is needed for far transfer to apply these skills to real-world cycling scenarios. This includes navigating turns and obstacles on a cycling track, as well as maintaining control over their riding path when encountering other riders.”

Point raised by Reviewer #2 C22

346-348 - The fear of falling is identified in literature as one of the biggest barriers to learning to cycle on a traditional bicycle, strengthen this sentence with literature references.

Response by author 

We tried to strengthen this sentence as,

“As noted in a previous systematic review [9], fear of falling is the biggest barrier to learning to cycle. However, our recent study found that most participants were able to reduce their fear of falling by regaining balance on the balance bike through the use of their feet and having greater control over their riding speed. This suggests that balance bike training can be effective method for reducing fear of falling not just for children and youth riders, but also for adult riders.”

Point raised by Reviewer #2 C23

358 – “Real-world situations”? Did the programme include a component of cycling on the road with car traffic?

Response by author 

In this study, the cycling scenario focused on the real-world situations on cycling track in Hong Kong, we modified the sentences to clarify it in the discussion.

“…However, it is important to recognize the potential influence of cultural differences on cycling behaviour. The recent design of the balance bike training programme focused on simulating cycling scenario on cycling tracks in Hong Kong, including the position of slow and fast lanes, cycling rules, and cycling etiquette. While this training was effective in improving participants’ cycling confidence in these scenarios, future studies in other settings should consider necessary amendments to account for cultural differences and the unique challenges of cycling in different scenarios, like cycling on road. By doing so, participants may be better able to develop cycling confidence needed to navigate a variety of cycling environments.”

Point raised by Reviewer #2 C24

385 – Limitations - Physical activity influences learning to cycle in children (there is at least one recent study that finds this association), can you strengthen this presentation with references. 

Response by author 

Thanks for your comments. While we acknowledge the importance of physical activity in learning to cycle, we have decided to keep our focus on the limitations of study design and small sample size. 

Point raised by Reviewer #2 C25

Another major limitation was not controlling for participants' prior experience on various bicycles. We do not know if the previous experience influenced the type of transfer found "quick positive motor skill transfer" "late positive motor skill transfer" and "zero transfer", however, considering that learning can be seen as the sum of experiences, we can assume that yes. This weakness should be presented and discussed.

Response by author 

Our original thought is to use a self-designed time frame to define “the speed of learning in transfer of learning”. However, we understand that this definition does not have sufficient evidence to support, therefore, we decided to report the time intervals of successful transfer only.

“Cycle Independently

… The time taken by participants to cycle independently was recorded and reported, in 15-minute intervals: below 15 minutes, 16 to 30 minutes, 30 to 45 minutes, and 45 to 60 minutes. The trial had a maximum time limit of 60 minutes.”

“Results

Phase 2: Assessment for balance bike Intervention

… Out of the eleven participants, eight were able to self-launch and ride a pedal bike continuously within 15 minutes, while the remaining three were able to cycle independently within 16 to 30 minutes (n = 1) and 45 to 60 minutes (n = 2).”

“Discussion

… Secondly, the lower threshold of MSE for transferring stability skills from a balance bike to a two-wheel bike was not established in our study, as all non-cyclists in this study were able to cycle after the balance bike training. Some participants even accumulated over 10.5 km of riding distance (i.e., high MSE) after only six sessions. If a future study could recruit a large sample size with varied MSE and ability to gain cycling skills, more information could be discovered to shorten the training duration. Thirdly, the lack of strict control over the physical activity background of the participants in this study may have contributed to the heterogeneity of participants’ backgrounds. Future studies should measure and control physical activity levels to include a more diverse population and make the findings more generalizable.”

Point raised by Reviewer #2 C26

392 – “Extreme evidence” based on a study of only 11 adults is abusive. please reconsider the use of this expression.

Response by author 

We apologize this inaccurate expression and revised to,

“This study provides valuable evidence for the effectiveness of balance bike tests and a novel balance bike intervention in teaching adults how to ride a bike… 

…this study provides valuable initial evidence and highlights the potential of the balance bike intervention as a useful tool for teaching cycling skills to adults.”

Point raised by Reviewer #2 C27

393 – How did you arrive at these 5 principles? This is a discussion to complement the discussion section.

Response by author 

We try to explain it in the results 

“Phase 1: Balance bike training programme development

The balance bike programme was based on five principles: riding speed, gliding to turn, cycling confidence, high MSE, and a dual-task approach…”

Reviewer #3

Point raised by Reviewer #3 C01

This manuscript proposes to study the transfer of skills between balance cycling and pedaling in adults.

The authors are to be congratulated for the methodological rigor they try to imprint in this study, however some questions arise as you read:

Response by author 

Thanks for the recognition and we are going to revise the manuscript to address the suggestions.

Point raised by Reviewer #3 C02

- Both the title and the second objective refer to transfer between motor skills, however, the authors do not describe what they mean by positive transfer or what is meant by transfer in general. It is recommended that this be done in the framework.

Response by author 

A paragraph has been revised and described in the Introduction section.

“The transfer of learning refers to the concepts that skills acquired during practice can be applied to other situations [11]. For example, if an individual learns to balance on a balance bike, they may be able to transfer that skill to riding a two-wheel bike. This is because the balance bike has helped them develop the necessary balance and coordination skills required to ride a two-wheel bike. Positive transfer occurs when the skills learned on the balance bike improve the individual's ability to a new but similar task, riding on a two-wheel bike [6]. On the other hand, negative transfer occurs if the skills learned on the bike with training wheels actually hinder the individual's ability to learn [6, 9]. Zero transfer would occur if the skills learned on a balance bike have no effect on their ability to ride a pedal bike.”

Point raised by Reviewer #3 C03

- Another aspect is that it is not made clear, which skill is meant to assess transfer. The authors should be clear in this definition, since the act of pedaling is not present in the first bicycle. This should be clarified.

Response by author 

Sentences had been added to clarify the concern,

“…if an individual learns to balance on a balance bike, they may be able to transfer that skill to riding a two-wheel bike. This is because the balance bike has helped them develop the necessary balance and coordination skills required to ride a two-wheel bike.”

Point raised by Reviewer #3 C04

In the Methods section - The bicycle being the means to test the transfer, the difference between the bicycles used for each condition is not explained.

Response by author 

Sentences in “Cycle Independently” were revised,

“At the end of the balance bike training programme, participants were invited to ride a 20-inch conventional pedal bike, which was of the same size as the balance bike used in the training.”

Point raised by Reviewer #3 C05

- In the Methods section, ln. 117-118 - the characteristics of each phase are presented. In phase 2, what is the rationale for choosing 8x20-minutes? Why not more or less time or sessions?

Response by author 

We discussed the rationale and benefits of our programme design in “Discussion” section

“Compared to previously published cycling training programmes that range from 135 [32] to 270 minutes [5], our study took a mid-range approach, with an accumulated training time of 160 minutes. This duration is relatively short, which is important since lack of time is a commonly reported barrier to engaging in structured exercise [33]. Accumulating multiple short bouts of exercise has been shown to provide similar health and fitness benefits as longer continuous exercise modes [34, 35]. Additionally, short bouts of exercise provide more opportunities for individuals to fit exercise into a busy day. Our recent multiple-short bouts design allowed participants to have a time-efficient programme that could potentially maintain high motivation and reduce the accumulation of physical and mental fatigue and burden on the participants.”

Point raised by Reviewer #3 C06

- In the Methods section, ln. 142 - anthropometric measurements were performed. What measurement protocols were used? 

Response by author 

A paragraph was added to clarify the information.

“Anthropometrics 

 Participants’ body height and weight were measured by a portable height rod stadiometer (Detecto, PHR, MO, USA) and digital scale (Tanita BWB-800, MA, USA) respectively. They were asked to stand on the height rod stadiometer and digital scale for the measurements. The graduation unit was 0.1 cm for height and 0.1 kg for weight.”

Point raised by Reviewer #3 C07

What expertise does the measurer have?

Response by author 

To clarify the information, sentences in Methods were revised. 

“The confidentiality of participants’ information Participants’ was ensured both during and after the data collection process. Additionally, all measurements were carried out by an experienced sports scientist, with the support of a research assistant in an indoor multi-game court to ensure accuracy and consistency. The STROBE checklist is available to the reader as S2.”

Point raised by Reviewer #3 C08

- In the Methods section, ln. 146 - the cultural context in motor learning is very important, as well as the learning possibilities between sexes, decades of birth and their motor availability. This difference between elements of each sex is a weakness of the study. To bridge these differences, assessing motor competence would be recommended. Did you do this? How do you support the results in light of the contexts?

Response by author 

In this study, we use CMJ test to assess physical functional capacity in our recent adult population. We found no difference on CMJ jump height between two groups in Phase 1, and we suggested that it is not related to the cycling capacity. 

“Large differences in effect size have been observed in both straight and two-curved turn sprints, but lower limb muscle-related performance on the CMJ test is not found to be related to cycling capabilities. While a systematic review and meta-analysis have shown correlations between measures of balance and lower-extremity muscle strength/power in healthy individuals across the lifespan … may not fully reflect the unique balance strategy and control required for cycling.”

Point raised by Reviewer #3 C09

- In the Methods section, ln. 166 - "Challenge-Taking Course." A schematic/image illustrating the course would be welcomed. This allows the principle of replicability to be guaranteed.

Response by author 

A figure was added. 

Fig. 3. A schematic image illustrating the common setting of the challenge course. 

Point raised by Reviewer #3 C10

- In the Methods section, ln. 172 - "Solve real-world challenges": Which ones? How were they simulated in this situation?

Response by author 

We made clarification and examples to make the description clear.

“Results

Phase 1: Balance bike training programme development

To challenge participants’ cycling stability, the training programme included activities that involved riding a balance bike while performing additional tasks (i.e., dual-task approach) that required executive functions [25], such as visual scanning to find a moving target or delivering an object with one hand when riding. The programme was designed as a series of challenges that simulated different scenarios encountered on cycling tracks, such as navigating traffic chaos or taking turns leading the group, to encourage participants to adapt to unexpected situations for better cycling stability.”

Point raised by Reviewer #3 C11

- In the Methods section, ln. 201-202 - There are several ways to calculate the optimal saddle height. The way they use is unclear, especially the "slightly bent" of the knees and hips. Can they be clearer? Have you not used any protocols?

Response by author 

Description was revised and reference was added.

“To ensure that participants could comfortably and safely balance on the bike during testing, they were ample time to become familiar with a 20-inch balance bike and the testing setup. The seat height of the bike was then adjusted to fit each participant’s body, enabling them to place both feet on the ground with slightly bent hips and knees [12, 17]. This position allowed for proper extension of the hip and knee joints when propelling themselves forward.”

Point raised by Reviewer #3 C12

- In the Methods section, ln. 219 - Is the classification "Quick positive transfer" or "late positive transfer" your classification or used by other authors? What is the rationale for having 15 minutes as the cut-off value? How does a person who can perform the task in 15 minutes differ so much from one who can do it in 16 minutes?

Response by author 

Our original thought is to use a self-designed time frame to define “the speed of learning in transfer of learning”. However, we understand that this definition does not have sufficient evidence to support, therefore, we decided to report the time intervals of successful transfer only.

“Cycle Independently

… The time taken by participants to cycle independently was recorded and reported, in 15-minute intervals: below 15 minutes, 16 to 30 minutes, 30 to 45 minutes, and 45 to 60 minutes. The trial had a maximum time limit of 60 minutes.”

“Results

Phase 2: Assessment for balance bike Intervention

… Out of the eleven participants, eight were able to self-launch and ride a pedal bike continuously within 15 minutes, while the remaining three were able to cycle independently within 16 to 30 minutes (n = 1) and 45 to 60 minutes (n = 2).”

Point raised by Reviewer #3 C13

- In the Results section, ln. 266 - The authors indicate that confidence in cycling was assessed across sessions. 

What results come from this...is there a session that shows a jump in confidence in bicycling? 

I recommend statistical treatment of these data.

Response by author 

To clarify the methods for cycling confidence, the sentences in “Cycling Confidence” were revised.

“Previous studies have used self-report cycling confidence to examine the effectiveness of cycling training among adults [19, 20]. In this study, cycling confidence was assessed verbally after each of the eight training sessions and the two testing sessions (i.e., T1 & T2), resulting in a total of 10 recorded confidence scores. Participants were asked to rate their confidence to ride a pedal bike on a five-point Likert scale (1 = not confident; 2 = slightly confident; 3 = somewhat confident; 4 = fairly confident; 5 = very confident).”

Accepted the recommendation to make a statistical treatment to determine the change of cycling confidence before and after the training programme. Therefore, new sentences were added in the data analysis and results.

“A Bayes factor for related-sample t test was used to determine the changes of cycling confidence at post-test (T2) compared to the pre-test level (T1).”

“According to the verbal survey conducted during Phase 2 of the study, participants reported feeling fairly to very confident with a score of cycling confidence: 4.45 ± 0.52 about riding a pedal bike after all training sessions. At the end of the study (T2), there was a significant increase in participants’ cycling confidence compared to that at T1 with a difference of 3.09 ± 0.83 (BF01 <0.001, p < 0.001, d = 3.72). Out of the eleven participants, eight were able to self-launch and ride a pedal bike continuously within 15 minutes, while the remaining three were able to cycle independently within 16 to 30 minutes (n = 1) and 45 to 60 minutes (n = 2).”

Point raised by Reviewer #3 C14

- In the Results section, ln. 303 - Was stability assessed in any way across sessions? No reference to such an evaluation appears, which makes the use of this term abusive.

Response by author 

In that line (original line 303), it is a section for Discussion. The sentence is to highlight that cycling is a standalone motor skill that is derived from fundamental skill.

---

## [Decision Letter · Decision Letter 1]

24 Nov 2023

PONE-D-23-13359R1

Positive Skill Transfer in Balance and Speed Control from Balance Bike to Pedal Bike in Adults: A Multiphase Intervention Study

PLOS ONE

Dear Dr. Chow,

Thank you for submitting your manuscript to PLOS ONE. After careful consideration, we feel that it has merit but does not fully meet PLOS ONE’s publication criteria as it currently stands. Therefore, we invite you to submit a revised version of the manuscript that addresses the points raised during the review process.

We look forward to receiving your revised manuscript.

Kind regards,

Bojan Masanovic, Ph.D.

Academic Editor

PLOS ONE

Journal Requirements:

Reviewers' comments:

Reviewer's Responses to Questions

**Comments to the Author**

1. If the authors have adequately addressed your comments raised in a previous round of review and you feel that this manuscript is now acceptable for publication, you may indicate that here to bypass the “Comments to the Author” section, enter your conflict of interest statement in the “Confidential to Editor” section, and submit your "Accept" recommendation.

Reviewer #1: (No Response)

Reviewer #4: All comments have been addressed

Reviewer #5: (No Response)

2. Is the manuscript technically sound, and do the data support the conclusions?

Reviewer #1: Yes

Reviewer #4: Yes

Reviewer #5: Yes

3. Has the statistical analysis been performed appropriately and rigorously? 

Reviewer #1: Yes

Reviewer #4: Yes

Reviewer #5: Yes

4. Have the authors made all data underlying the findings in their manuscript fully available?

Reviewer #1: Yes

Reviewer #4: Yes

Reviewer #5: Yes

5. Is the manuscript presented in an intelligible fashion and written in standard English?

Reviewer #1: No

Reviewer #4: Yes

Reviewer #5: Yes

6. Review Comments to the Author

Reviewer #1: The authors addressed many concerns adequately. However, some issues remain insufficiently addressed. Further proof reading is needed.

Abstract –

Avoid inflated language like “extreme” evidence, let the data speak for the impact.

Introduction – I suggest simplifying the background to be more directly related to your study purpose and outcomes. For example, you provide information about parents teaching children to ride, however, this is not a purpose or outcome of your study, so it is not relevant.

Also, is there evidence to suggest that increased cycle track miles have results in earlier learning. I do not see evidence of a cause-and-effect relationship.

Line 84, be more specific of what challenges balance bikes have helped to overcome. It does not seem like they overcome all the challenges listed before this statement.

Introduction can be examined to improve flow

Reviewer #4: I think that this study is original and that the results can contribute to practical application, and given results could be used for future reasrch on the same topic

Reviewer #5: very interesting study, and very current, considering that it includes elderly people.

A couple of suggestions:

- the tables should be formatted exactly the same way (table 1 and the same table)

- the title of the table should be (in one table it is bold, and in the other table it is not)

- Hypotheses are mentioned in the Discussion (paragraph 381-385), but I did not notice that the hypotheses were mentioned in the Method.

7. PLOS authors have the option to publish the peer review history of their article (what does this mean?). If published, this will include your full peer review and any attached files.

Reviewer #1: No

Reviewer #4: No

Reviewer #5: No

---

## [Author Response · Author response to Decision Letter 1]

27 Nov 2023

Reviewer #1 - Comment #1: 

The authors addressed many concerns adequately. However, some issues remain insufficiently addressed. Further proof reading is needed.

Response to Reviewer #1 - Comment #1:

We made further proof reading and provided our best effort to address the concerns. Please find the point-to-point response below:

---

Reviewer #1 - Comment #2: 

Abstract –

Avoid inflated language like “extreme” evidence, let the data speak for the impact.

Response to Reviewer #1 - Comment #2:

The abstract has been updated according to the comment, we agree to “let the data speak the impact”.

Line 37 – 40 “…with Bayes factor analyses providing evidence of this difference, BF01 = 0.228 in the 15 m sprint test and BF01 = 0.138 in the two-turn curved sprint test. The novel training programme in Phase 2 remarkable effectiveness in improving their balance bike riding performance…”

Line 44 – 45 “…It provides extreme evidence that healthy adults can learn to ride independently with the help of a balance bike, a tool that's commonly used for teaching children…”

Besides, to clarify the interpretation of Bayes Factor, a paragraph in data analysis has been updated and a updated reference was added.

Page 14, line 286 – 292 

“The strength of the evidence for the null hypothesis against the alternative hypothesis (or models) can be expressed as a Bayes Factor (BF01), which is an odds ratio. The size of BFs can be interpreted as providing anecdotal (BF01 = 1/3–1), moderate (BF01 = 1/10–1/3), strong (BF01 = 1/30–1/10), very strong (BF01 = 1/100–1/30), and extreme (BF01 < 1/100), in favour of alternative hypothesis compared to null hypothesis [2320, 21]. Evidence for the null hypothesis was set as anecdotal (BF01 = 1–3), moderate (BF01 = 3–10), strong (BF01 = 10–30), very strong (BF01 = 30–100), and extreme (BF01 > 100) [20, 21].”

New reference:

Kelter R. “Bayesian alternatives to null hypothesis significance testing in biomedical research: A non-technical introduction to Bayesian inference with JASP. BMC Med. Res. Methodol. 2020;20(1): 142, https://doi.org/10.1186/s12874-020-00980-6.

---

Reviewer #1 - Comment #3: 

Introduction – I suggest simplifying the background to be more directly related to your study purpose and outcomes. For example, you provide information about parents teaching children to ride, however, this is not a purpose or outcome of your study, so it is not relevant.

Also, is there evidence to suggest that increased cycle track miles have results in earlier learning. I do not see evidence of a cause-and-effect relationship.

Response to Reviewer #1 - Comment #3:

We agree to simplify the introduction.

Line 62 – 72 

“…Learning to cycle independently is a common rite of passage for children… potentially delaying the age at which individuals learn to cycle independently [3].”

Therefore, the reference list is also updated.

Page 30 – 35 

---

Reviewer #1 - Comment #4: 

Line 84, be more specific of what challenges balance bikes have helped to overcome. It does not seem like they overcome all the challenges listed before this statement.

Introduction can be examined to improve flow

Response to Reviewer #1 - Comment #4:

Thanks for the comments. We revised the paragraph and clarified what challenges were. 

Line 73 

“In recent times, balance bikes have emerged as a popular alternative to overcome the challenges associated with traditional learning approaches for cycling. These challenges primarily involve ineffective methods and negative experiences. Balance bikes, which lack pedals and allows rider to walk or sprint while seated [5], offer a more intuitive and effective way to acquire skills and improve cycling stability. In fact, studies have shown that balance bikes are more effective in helping riders learn the essential skill of maintaining balance on two wheels compared to bikes with training wheels, which prioritize on pedalling skills [3, 4]. With the right support, including access to affordable, appropriately sized adult balance bikes, along with effective learning design, individuals of all ages can overcome these barriers and learn to cycle independently.”

---

Reviewer #4 - Comment #1: 

I think that this study is original and that the results can contribute to practical application, and given results could be used for future reasrch on the same topic

Response to Reviewer #4 - Comment #1:

Thanks for the positive comments.

---

Reviewer #5 - Comment #1: 

very interesting study, and very current, considering that it includes elderly people.

Response to Reviewer #5 - Comment #1:

Thanks for the positive comments.

---

Reviewer #5 - Comment #2: 

A couple of suggestions:

- the tables should be formatted exactly the same way (table 1 and the same table)

- the title of the table should be (in one table it is bold, and in the other table it is not)

Response to Reviewer #5 - Comment #2:

Table 1 and 2 had been updated. 

Page 15, line 286

Page 16, line 297 – Page18, line 301 

---

Reviewer #5 - Comment #3: 

- Hypotheses are mentioned in the Discussion (paragraph 381-385), but I did not notice that the hypotheses were mentioned in the Method.

Response to Reviewer #5 - Comment #3:

The hypotheses were added to the Method.

Page 9, line 172 – 179

“Based on the available literature, the following hypotheses have been derived concerning cycling experience and the effectiveness of the balance bike training intervention: Hypothesis 1: It is anticipated that individuals with prior cycling experience, referred to as cyclists, will display superior performance in riding and controlling a balance bike compared to individuals without prior cycling experience, referred to as non-cyclists. Hypothesis 2: Non-cyclists who undergo the balance bike training intervention will demonstrate enhanced performance in balance bike tests and experience an increase in confidence levels when riding a two-wheel bike after engaging in practice with a balance bike.”

---

## [Decision Letter · Decision Letter 2]

21 Jan 2024

Positive Skill Transfer in Balance and Speed Control from Balance Bike to Pedal Bike in Adults: A Multiphase Intervention Study

PONE-D-23-13359R2

Dear Dr. Chi Ching, Gary Chow,

We’re pleased to inform you that your manuscript has been judged scientifically suitable for publication and will be formally accepted for publication once it meets all outstanding technical requirements.

Kind regards,

Bojan Masanovic, Ph.D.

Academic Editor

PLOS ONE

Additional Editor Comments (optional):

The rematch lasted several laps.

It was difficult to bring it to an end.

Some reviewers did not respond later.

The authors have worked diligently and I think that in the end they deserve to have the manuscript published.

I think that the quality of this manuscript has reached the level of publication in Plos One journal.

Reviewers' comments:

Reviewer's Responses to Questions

**Comments to the Author**

1. If the authors have adequately addressed your comments raised in a previous round of review and you feel that this manuscript is now acceptable for publication, you may indicate that here to bypass the “Comments to the Author” section, enter your conflict of interest statement in the “Confidential to Editor” section, and submit your "Accept" recommendation.

Reviewer #1: All comments have been addressed

Reviewer #5: (No Response)

2. Is the manuscript technically sound, and do the data support the conclusions?

Reviewer #1: Yes

Reviewer #5: Yes

3. Has the statistical analysis been performed appropriately and rigorously? 

Reviewer #1: Yes

Reviewer #5: Yes

4. Have the authors made all data underlying the findings in their manuscript fully available?

Reviewer #1: Yes

Reviewer #5: Yes

5. Is the manuscript presented in an intelligible fashion and written in standard English?

Reviewer #1: Yes

Reviewer #5: Yes

6. Review Comments to the Author

Reviewer #1: The authors have done a nice job addressing remaining comments from all reviewers. The article addresses an important gap in the literature.

Reviewer #5: The topic is very interesting. Also, the work can serve as motivation for certain study programs and curricula. In addition, this scientific work is methodologically well-conceived and appropriate statistical procedures were used. In the chapter provided for the methodology, the methods of research implementation are explained in detail. The results of the research are clearly presented in tables. The discussion could have been more extensive, given the scope of the work, but this is not a crucial complaint.

In the work, the authors consulted contemporary and adequate literature, including 75 library units.

After the corrections, I am satisfied with the final version of the paper.

7. PLOS authors have the option to publish the peer review history of their article (what does this mean?). If published, this will include your full peer review and any attached files.

Reviewer #1: No

Reviewer #5: No

---

## [Editor Report · Acceptance letter]

21 Feb 2024

PONE-D-23-13359R2 

PLOS ONE

Dear Dr. Chow, 

I'm pleased to inform you that your manuscript has been deemed suitable for publication in PLOS ONE. Congratulations! Your manuscript is now being handed over to our production team.

Kind regards, 

on behalf of

Dr. Bojan Masanovic 

Academic Editor

PLOS ONE